# ZIGZAG DIFFUSION SAMPLING: DIFFUSION MODELS CAN SELF-IMPROVE VIA SELF-REFLECTION

**Lichen Bai**[1]  **Shitong Shao**[1]  **Zikai Zhou**[1]  **Zipeng Qi**[2]
**Zhiqiang Xu**[3]  **Haoyi Xiong**[4]  **Zeke Xie**[1‡]
[1]xLeaF Lab, The Hong Kong University of Science and Technology (Guangzhou)
[2]Beihang University  [3]Mohamed bin Zayed University of Artificial Intelligence  [4]Baidu Inc
{lichenbai, sshao213, zikaizhou, zekexie}@hkust-gz.edu.cn

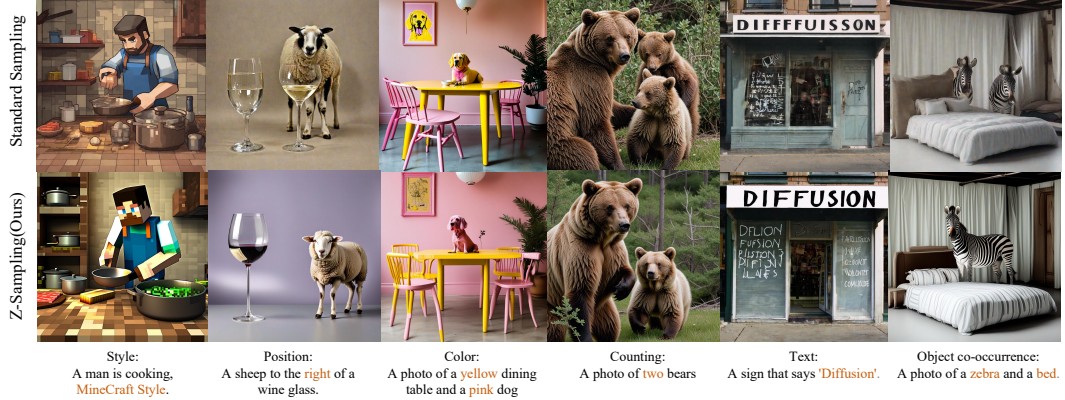

Figure 1: The qualitative results of Z-Sampling demonstrate the effectiveness of our method in various aspects, such as style, position, color, counting, text rendering, and object co-occurrence. We present more cases in Appendix D.2.

## ABSTRACT

Diffusion models, the most popular generative paradigm so far, can inject conditional information into the generation path to guide the latent towards desired directions. However, existing text-to-image diffusion models often fail to maintain high image quality and high prompt-image alignment for those challenging prompts. To mitigate this issue and enhance existing pretrained diffusion models, we mainly made three contributions in this paper. First, we propose *diffusion self-reflection* that alternately performs denoising and inversion and demonstrate that such diffusion self-reflection can leverage the guidance gap between denoising and inversion to capture prompt-related semantic information with theoretical and empirical evidence. Second, motivated by theoretical analysis, we derive Zigzag Diffusion Sampling (Z-Sampling), a novel self-reflection-based diffusion sampling method that leverages the guidance gap between denosing and inversion to accumulate semantic information step by step along the sampling path, leading to improved sampling results. Moreover, as a plug-and-play method, Z-Sampling can be generally applied to various diffusion models (e.g., accelerated ones and Transformer-based ones) with very limited coding and computational costs. Third, our extensive experiments demonstrate that Z-Sampling can generally and significantly enhance generation quality across various benchmark datasets, diffusion models, and performance evaluation metrics. For example, DreamShaper with Z-Sampling can self-improve with the HPSv2 winning rate up to **94%** over the original results. Moreover, Z-Sampling can further enhance existing diffusion models combined with other orthogonal methods, including Diffusion-DPO. The code is publicly available at github.com/xie-lab-ml/Zigzag-Diffusion-Sampling.

---

‡Corresponding author

## 1   INTRODUCTION

Diffusion models, known for its powerful generative capabilities and diversity, have become a mainstream generation paradigm in images (Podell et al., 2023; Lin et al., 2024b; Qi et al., 2025), videos (Ho et al., 2022; Blattmann et al., 2023), and 3D objects (Luo & Hu, 2021; Voleti et al., 2024) and beyond. One key ability of diffusion models is to guide the sampling path based on some conditions (e.g., texts), leading to conditional or controllable generation (Ho & Salimans, 2022).

However, while strong guidance may improve semantic alignment to those challenging prompts, it often causes significant decline in image fidelity, leading to mode collapse, and resulting inevitable accumulation of errors during the sampling process (Chung et al., 2024). To mitigate this issue, some studies apply additional manifold constraints to the sampling paths (Chung et al., 2024; Yang et al.; He et al.), which compromises the diversity of generated outputs. Others design varying guidance scales across different denoising regions to mitigate this issue (Shen et al., 2024), but such explicit strategies often lead to unnatural outputs. Thus, enhancing high generation quality while maintaining prompt alignment effectively during sampling remains a crucial challenge, especially for those challenging prompts. This challenge may require more controllable prompt guidance beyond classical guidance like classifer-free guidance (Ho & Salimans, 2022).

Fortunately, we discover that semantic information may be inherently embedded in the random latent space, influencing the quality of image generation (Xu et al., 2024b; Po-Yuan et al., 2023; Mao et al., 2023b; Wu et al., 2023c). In Figure 2, we demonstrate the following phenomenon: if a latent can generate images aligned with a specific concept $c$ under no conditional prompt, it will generate high-quality results with $c$ as the conditional prompt. This implies that the latent naturally carries relevant semantic information and can align with relevant semantic prompts very well. Figure 3 intuitively illustrates that the green initial point with certain semantic information is usually superior to the red initial point for the prompts associated with the semantic information.

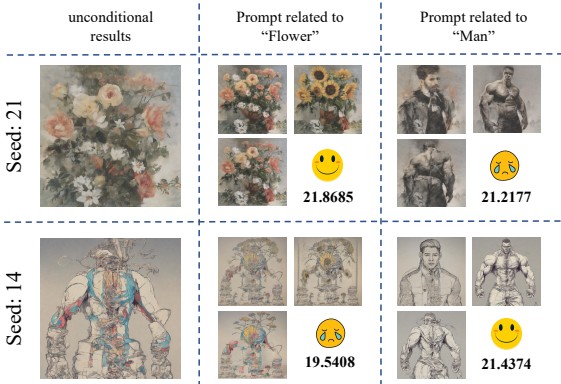

Figure 2: Semantic-rich latents effectively generate images aligned with intended semantics. For instance, the random latent (seed 21) is better suited for generating images related to the concept of "flowers". We present more cases in Appendix C.1.

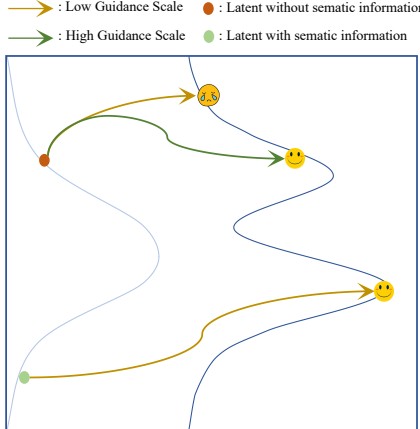

Figure 3: If the latent carries semantic information, we can obtain prompt-related results from this latent even without conditional guidance.

We fortunately discover that employing strong guidance during denoising process and weak guidance during inversion process establishes a guidance gap between denoising and inversion that can inject prompt semantic information to the latent. We present more examples and discussion in Appendix C.2. Can this insight lead to improved sampling methods? We note that large language models (LLMs) can self-improve reasoning through self-reflection (Ji et al., 2023; Shinn et al., 2024). However, the self-reflection mechanism that can self-improve diffusion sampling has not been reported in previous studies.

Motivated by our observation and self-reflection in LLMs, to the best of our knowledge, we are the first to formulate diffusion self-reflection that let a latent denoise in a zigzag manner, namely a denoising step and an inversion step alternately, step by step along the sampling path. As Figure 4

illustrates, we propose Zigzag Diffusion Sampling, or Z-Sampling, which can capture semantic information with such repeated zigzag self-reflection operations and move to more desirable results along the sampling path. Through each zigzag self-reflection operation, the latent accumulates more semantic information.

The contributions of this work can be summarized as follows.

First, we theoretically and empirically demonstrate that the guidance gap between denoising and inversion of diffusion self-reflection can capture the semantic information embedded in the latent space, which matters to generation quality and prompt-image alignment.

Second, motivated by the theoretical results, we derive Z-Sampling, a novel self-reflection-based diffusion sampling method that can leverage the guidance gap to accumulate semantic information through each zigzag self-reflection step and generate more desirable results. It allows flexible control over the injection of semantic information and is applicable across various diffusion architectures with very limited coding costs.

Third, extensive experiments demonstrate the effectiveness and generalization of Z-Sampling across various benchmark datasets, diffusion models, and evaluation metrics. As theoretical analysis suggests, diffusion models with Z-Sampling especially self-improve for challenging complex or fine-grained prompts, such as position, counting, color attribution, and multi-object, breaking through the performance peak of pretrained diffusion models. Moreover, orthogonal methods, such as Diffusion-DPO (Wallace et al., 2024), can further self-improve with Z-Sampling. Importantly, as a training-free method, Z-Sampling can still exhibit significant improvements over the baselines with limited computational cost, which suggests its efficiency and practical value. In the efficiency study, even with 36% less computational time, Z-Sampling can reach the best performance of standard sampling.

---

**Algorithm 1** Z-Sampling

1: **Input:** Denoising at timestep $t$: $\Phi^t$, Inversion at timestep $t$: $\Psi^t$, text prompt: $c$, denoising guidance: $\gamma_1$, inversion guidance: $\gamma_2$, inference steps: $T$, zigzag optimization steps: $\lambda$
2: **Output:** Clean image $x_0$
3: Sample Gaussian noise $x_T$
4: **for** $t = T$ **to** 1 **do**
5:     $x_{t-1} = \Phi^t(x_t|c, \gamma_1)$
6:     **if** $t > T - \lambda$ **then**
7:         #Inversion by equation 4
        $\tilde{x}_t = \Psi^t(x_{t-1}|c, \gamma_2)$
8:         #Denoising by equation 2
        $x_{t-1} = \Phi^t(\tilde{x}_t|c, \gamma_1)$
9:     **end if**
10: **end for**
11: **return** $x_0$

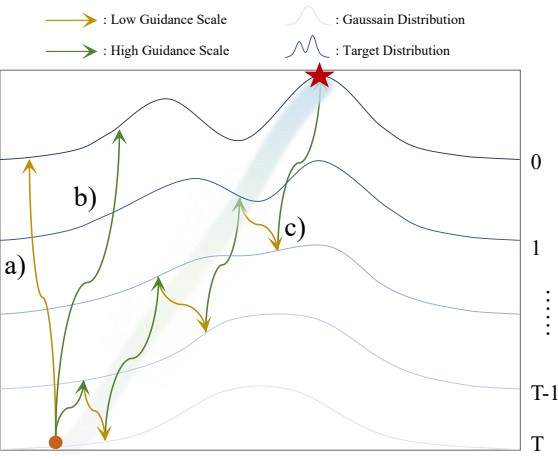

Figure 4: The illustration of our method: a) weak guidance sampling; b) strong guidance sampling; c) Z-Sampling (with diffusion self-reflection).

## 2 PRELIMINARIES

In this section, we formally introduce prerequisites and background.

**Diffusion Model.** We define the total numeber of denoising steps $T$ and conditional prompt $c$. Given the denoising procss $\Phi : \mathcal{N} \times \mathcal{C} \rightarrow \mathcal{D}$ and guidance scale $\gamma_1$, starting from $x_T \in \mathcal{N}$, we can generate $x_0 = \Phi(x_T|c, \gamma_1) \in \mathcal{D}$, where $\mathcal{N}$ represents the distribution of Gaussian and $\mathcal{D}$ represents the distribution of target data. We note that the mapping function $\Phi$ corresponds to the probability $P(x_0|c, \gamma_1, x_{1:T})$. For simplicity, we simplify only the initial input $x_T$ in $\Phi$. Similarly, we can also reverse this process, given the inversion process $\Psi : \mathcal{D} \times \mathcal{C} \rightarrow \mathcal{N}$ under guidance scale $\gamma_2$, we obtain inverted data $\tilde{x}_T = \Psi(\tilde{x}_0|c, \gamma_2) \in \mathcal{N}$ from $\tilde{x}_0 \in \mathcal{D}$.

Following Ho et al. (2020), we treat diffusion model as a Monte Carlo process and decompose $\Phi$ into $T$ times single-step denoising mappings as

$$\Phi(x_T|c, \gamma_1) = \Phi^T(x_T|c, \gamma_1) \circ \Phi^{T-1}(x_{T-1}|c, \gamma_1) \circ \cdots \circ \Phi^2(x_2|c, \gamma_1) \circ \Phi^1(x_1|c, \gamma_1). \quad (1)$$

And we define $\Phi^t$ as

$$x_{t-1} = \Phi^t(x_t|c, \gamma) = \sqrt{\alpha_{t-1}}\frac{x_t - \sqrt{1-\alpha_t}\epsilon_\theta^t(x_t)}{\sqrt{\alpha_t}} + \sqrt{1-\alpha_{t-1}}\epsilon_\theta^t(x_t), \quad (2)$$

where $a_t := \prod_{i=1}^t (1 - \beta_i)$ and $\beta_t$ are the pre-defined parameters for scheduling the scales of adding noises in DDIM scheduler (Song et al., 2020). we denote $\epsilon_\theta^t$ as the predicted score by the denoising network $\theta$ at timestep $t$, with further details provided in the next paragraph.

Similarly, for the inversion process $\Psi$, we can also perform this decomposition as

$$\Psi(\tilde{x}_0|c, \gamma_2) = \Psi^1(\tilde{x}_0|c, \gamma_2) \circ \Psi^2(\tilde{x}_1|c, \gamma_2) \circ \cdots \circ \Psi^{T-1}(\tilde{x}_{T-2}|c, \gamma_2) \circ \Psi^T(\tilde{x}_{T-1}|c, \gamma_2), \quad (3)$$

where we obtain $\tilde{x}_{t-1}$ via $\Psi^t$ as

$$\tilde{x}_t = \Psi^t(\tilde{x}_{t-1}|c, \gamma_2) = \sqrt{\frac{\alpha_t}{\alpha_{t-1}}}\tilde{x}_{t-1} + \sqrt{\alpha_t}\left(\sqrt{\frac{1}{\alpha_t}-1} - \sqrt{\frac{1}{\alpha_{t-1}}-1}\right)\epsilon_\theta^t(\tilde{x}_{t-1}). \quad (4)$$

In equation 4 we approximate the score predicted at timestep $t$ with timestep $t-1$ along the inversion path, i.e, set $\epsilon_\theta^t(\tilde{x}_{t-1}) \approx \epsilon_\theta^t(\tilde{x}_t)$. If this approximation error is negligible, $\Phi$ and $\Psi$ can be proven to be inverse functions (Mokady et al., 2023), meaning that $\Psi = \Phi^{-1}$.

**Classifier free guidance.** Controllable generation typically involves guiding or constraining the semantic representation. In classifier free guidance (Ho & Salimans, 2022), a score prediction network $u_\theta$ is trained both conditionally and unconditionally. During inference, denoising scores are computed by interpolating between conditional and unconditional scores predicted by $u_\theta$, thus enabling the adjustment of guidance scale across various levels.

Specifically, for denoising and inversion process, we use guidance scales $\gamma_1$ and $\gamma_2$, with the corresponding scores as

$$\left.\begin{array}{l} \epsilon_\theta^t(x_t) = (1+\gamma_1)u_\theta(x_t, c, t) - \gamma_1 u_\theta(x_t, \varnothing, t) \\ \epsilon_\theta^t(\tilde{x}_t) = (1+\gamma_2)u_\theta(\tilde{x}_t, c, t) - \gamma_2 u_\theta(\tilde{x}_t, \varnothing, t) \end{array}\right\}, \quad (5)$$

where $u_\theta$ is the noise predictor, and $\varnothing$ is the null prompt, representing the denoising result under unconditional settings.

## 3 METHODOLOGY

In this section, we discuss how to encode semantic information into latents through the guidance gap and derive Z-Sampling according to theoretical analysis.

### 3.1 LATENTS WITH RELEVANT SEMANTIC INFORMATION

Our inspiration stems from the question: what makes a good latent in the diffusion process? As Figure 3 illustrates, we argue that a latent with relevant semantic information (green point) can align with the prompt under weak or sometimes even negative conditional guidance. In contrast, a latent lacking semantic information (red point) necessitates strong conditional guidance to attain comparable alignment and may remain unaligned under unconditional generation.

To verify this, we generate images using different latents (seeds) under unconditional settings, shown in Figure 2. We observe that if a latent can generate a image of a certain concept $c$ unconditionally, then, under certain prompt guidance, this latent usually performs higher in generating images related to $c$ compared to other latents. For example, in Figure 2, if the latent (seed 21) generates the images of flowers unconditionally, it yields higher-quality images when used with flower-related prompts in conditional generation. Previous studies also argued that the properties of latents partially predetermine image composition or contents during generation, affecting object position, size, and depth (Wu et al., 2023c; Guttenberg, 2023; Lin et al., 2024a; Xu et al., 2024b; Mao et al., 2023b). However, they did not formally explore how to encode semantic information into the latents.

## 3.2 Capture semantic information from the guidance gap

Considering a denoising process $\Phi : \mathcal{N} \times \mathcal{C} \to \mathcal{D}$, under text condition $c \in \mathcal{C}$, we sample an initial latent $x_T \in \mathcal{N}$, and obtain the generated data $x_0$ as

$$x_0 = \Phi(x_T | c, \gamma_1), \tag{6}$$

where $\gamma_1$ is condition guidance scale during denoising. Now, we further perform inversion operation on $x_0$ under the guidance scale of $\gamma_2$ as

$$\tilde{x}_T = \Psi(x_0 | c, \gamma_2). \tag{7}$$

If the approximation error in the inversion process is negligible, meaning $\Psi^{-1} = \Phi$, then equation 7 can be equivalently inverted as

$$\tilde{x}_0 = \Psi^{-1}(\tilde{x}_T | c, \gamma_2) = \Phi(\tilde{x}_T | c, \gamma_2). \tag{8}$$

Generally, the denoising guidance scale $\gamma_1$ is set to a common value (e.g., $\gamma_1 = 5.5$) to maintain standard generation and alignment to the prompt (Ho & Salimans, 2022). Conversely, the inversion guidance scale $\gamma_2$ is usually set to a small value (e.g., $\gamma_2 = 0$) to achieve inversion with weak guidance (Mokady et al., 2023). By comparing equation 6 and equation 8, we note that starting from $\tilde{x}_T$, we can generate $x_0$ under weak or even unconditional guidance scale $\gamma_2 = 0$. In contrast, starting from $x_T$ requires strong conditional guidance scale $\gamma_1 = 5.5$ to produce similar results.

According to the insight discussed in Section 3.1, if a initial latent can generate results related to prompt $c$ under weak guidance, it indicates this latent contains more semantic information related to $c$. Since guidance scale $\gamma_2$ is less than $\gamma_1$, we argue that the corresponding inverted latent $\tilde{x}_T$ contains more semantic information compared to $x_T$. We present more empirical evidence in Appendix C.2,

## 3.3 Zigzag Diffusion Sampling

Now we know that the guidance gap can capture additional semantic information. The next question is how to effectively leverage this property to inject semantic information into the sampling process.

**Vanilla Inversion**    A vanilla way is to use the inverted latent $\tilde{x}_T$ in place of $x_T$ as the starting point to generate semantically aligned results in the denoising process (see Algorithm 2). We provide Theorem 1 and show that the difference between the original $x_T$ and the inverted $\tilde{x}_T$, namely $\delta_{end2end} = (x_T - \tilde{x}_T)^2$, may reveal how significant the vanilla end-to-end information injection is. An illustrative diagram of the latents' difference is provided in Figure 27 (a) of Appendix F.

**Theorem 1 (See the proof in Appendix F.1)** *For a random latent $x_T \in \mathcal{N}$ and an inverted latent $\tilde{x}_T$ given by equation 7, the latent difference $\delta_{end2end}$ between $x_T$ and $\tilde{x}_T$ is*

$$\delta_{end2end} = (x_T - \tilde{x}_T)^2 = \alpha_T (\sum_{t=1}^{T} h_t (\underbrace{\epsilon_\theta^t(x_t) - \epsilon_\theta^t(\tilde{x}_t)}_{\tau_1(t):\text{semantic information gain term}} + \underbrace{\epsilon_\theta^t(\tilde{x}_t) - \epsilon_\theta^t(\tilde{x}_{t-1})}_{\tau_2(t):\text{approx error term}}))^2, \tag{9}$$

*where $h_t = \sqrt{1/\alpha_t - 1} - \sqrt{1/\alpha_{t-1} - 1}$, and $\epsilon_\theta^t(\cdot)$ is the predicted score given by equation 5.*

Here, $\tau_1(t)$ represents the semantic information gain induced by the guidance gap at timestep $t$, whereas $\tau_2(t)$ represents the approximation error inherent in the inversion process, which may be neglected for semantic information. We note that in equation 9, the end-to-end aggregation may let the sum of the semantic information $\tau_1$ over each step be small and fail to accumulate the desired semantic information gain step-by-step.

**Z-Sampling**    To let $\tau_1$ of each step be accumulated step-by-step instead of being canceled out in the vanilla sum, we decompose $\Phi$ into $\{\Phi^1, \Phi^2, \cdots, \Phi^T\}$, as defined in equation 1. We first denoise $x_t$ to obtain $x_{t-1} = \Phi^t(x_t | c, \gamma_1)$ and then we invert $x_{t-1}$ to get $\tilde{x}_t = \Psi^t(x_{t-1} | c, \gamma_2)$ for each timestep $t \in [T, 1]$. We may call such zigzag denoising-and-inversion operation along the diffusion sampling path as *diffusion self-reflection*. The proposed Z-Sampling method is presented in Algorithm 1 and illustrated in Figure 4. Note that Z-Sampling injects semantic information by replacing $x_t$ with $\tilde{x}_t$ at each timestep. We prove Theorem 2 and demonstrate the cumulative latent difference $\delta_{Z-Sampling} = \sum_{t=1}^{T}(x_t - \tilde{x}_t)^2$, depicted in Figure 27 (b) of Appendix F.

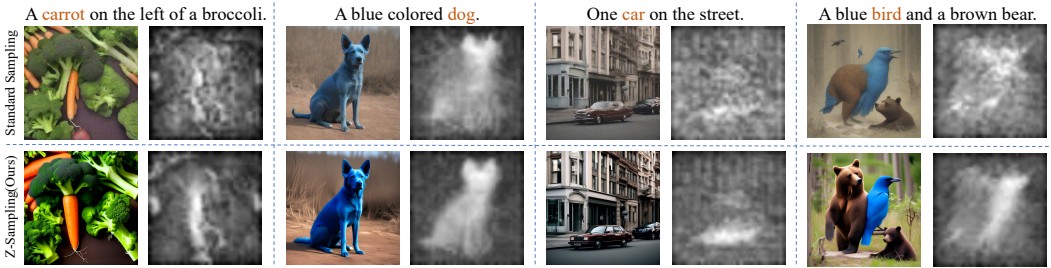

Figure 5: The cross-attention map highlights the interaction between the entity token (red color) and latent variables. Z-Sampling optimizes the latent so that it is more suitable for generating concepts in the related-prompt. For example, in the zigzag path of the second column, semantically injected latents exhibit sharper attention on "dog" with relatively clear boundaries.

**Theorem 2 (See the proof in Appendix F.2)** *Suppose $x_t$ is the denoised latent at step $t$, and $\tilde{x}_t$ be the corresponding inverted latent given by equation 4. Then the cumulative latent difference in Z-Sampling can be written as*

$$\delta_{\text{Z-Sampling}} = \sum_{t=1}^{T}(x_t - \tilde{x}_t)^2 = \sum_{t=1}^{T}\alpha_t h_t^2 (\underbrace{\epsilon_\theta^t(x_t) - \epsilon_\theta^t(\tilde{x}_t)}_{\tau_1(t):\text{semantic information gain term}} + \underbrace{\epsilon_\theta^t(\tilde{x}_t) - \epsilon_\theta^t(\tilde{x}_{t-1})}_{\tau_2(t):\text{approx error term}})^2, \quad (10)$$

*where $h_t$ and $\epsilon_\theta^t(\cdot)$ are consistent with Theorem 1.*

Again, focusing on the semantic information gain term, we report that $\delta_{end2end} \propto (\sum_1^T \tau_1(t))^2$ holds for vanilla inversion and $\delta_{Z-Sampling} \propto \sum_1^T (\tau_1(t))^2$ holds for Z-Sampling. Given the Jensen's inequality, we have $\sum_1^T (\tau_1(t))^2 \geq (\sum_1^T \tau_1(t))^2$, showing that the cumulative semantic information gain $\delta_{\text{Z-Sampling}}$ is larger than the end-to-end semantic information gain $\delta_{\text{end2end}}$. The semantic information gain induced by the guidance gap in Z-Sampling can be effectively accumulated, solving the previous issue of the semantic information gain cancellation.

We further prove Theorem 3 and show the significant impact of the guidance gap $\delta_\gamma$ on $\delta_{Z-Sampling}$.

**Theorem 3 (See the proof in Appendix F.3)** *Under the conditions of Theorem 2, the cumulative semantic information gain in Z-Sampling can be written as*

$$\delta_{\text{Z-Sampling}} = \sum_{t=1}^{T}\alpha_t h_t^2 (\delta_\gamma (u_\theta(x_t, c, t) - u_\theta(x_t, \varnothing, t)))^2, \quad (11)$$

*where the guidance gap is defined as $\delta_\gamma = \gamma_1 - \gamma_2$.*

We note that the larger $\delta_\gamma$, the more pronounced the effect of Z-Sampling. When $\delta_\gamma = 0$, it is approximately equivalent to standard sampling. This is also empirically verified in Figure 8.

In Figure 5, we visualize the cross-attention map of Z-Sampling during the early stages (i.e, $t/T = 49/50$) of the generation process. We observe that Z-Sampling indeed makes the attention regions corresponding to entity tokens more semantically focused, further illustrating the effectiveness of Z-Sampling on the semantic information gain. Mao et al. (2023b) reported that certain regions in random latents can induce objects representing specific concepts, which aligns with our observation that Z-Sampling enhances the association of certain regions with the prompt. Additionally, we discuss the impact of the approximation error $\tau_2$ in Appendix E.2 and E.3.

# 4 EMPIRICAL ANALYSIS

In this section, we conduct extensive experiments to demonstrate the effectiveness of our method.

## 4.1 EXPERIMENTS SETTING

**Datasets**  Pick-a-Pic (Kirstain et al., 2023), DrawBench dataset (Saharia et al., 2022), and GenEval (Ghosh et al., 2024). We leave more details in Appendix A.1.

**Metrics** We use multiple evaluation metrics, including HPS v2 (Wu et al., 2023c), PickScore (Kirstain et al., 2023), and ImageReward (IR) (Xu et al., 2024a). They are trained on large-scale human preference datasets, providing a reliable indication of genuine human preferences. Furthermore, we also employ the traditional metric AES (Schuhmann et al., 2022), which purely evaluate image quality. More details are found in Appendix A.2.

**Diffusion Models** We use various diffusion models as the generation backbone in main experiments. For SD2.1 (Rombach et al., 2022), SDXL (Podell et al., 2023), and Hunyuan-DiT (Li et al., 2024), we perform 50 denoising steps. For DreamShaper-xl-v2-turbo, which achieves efficient and high-quality generation by fine-tuning SDXL Turbo (Sauer et al., 2023), we set denoising step $T$ only to 4. And we set $\gamma_1 = 5.5$ in SDXL/SD2.1, $\gamma_1 = 6.0$ in Hunyuan-DiT, and $\gamma_1 = 3.5$ in DreamShaper-xl-v2-turbo, all to the recommended default values. We set the zigzag operation to be executed throughout the entire path ($\lambda = T - 1$) and inversion guidance scale $\gamma_2$ as zero, unless we specify them otherwisely.

**Baselines** We validate the effectiveness of Z-Sampling and compare it against the following baselines: **(a) standard sampling**, we first select four models: SD-2.1, SDXL, Hunyuan-DiT, and DreamShaper-xl-v2-turbo; **(b) resampling** (lug, 2022), repeatedly performs denoising at the same timestep by adding random noise to maintain the latent on the data manifold. Due to the page limit, other baselines such as AYS Sampling (Sabour et al., 2024), Diffusion DPO (Wallace et al., 2024), SEG (Hong, 2024), and CFG++ (Chung et al., 2024) are discussed in Appendix A.3.

## 4.2 MAIN EXPERIMENTS

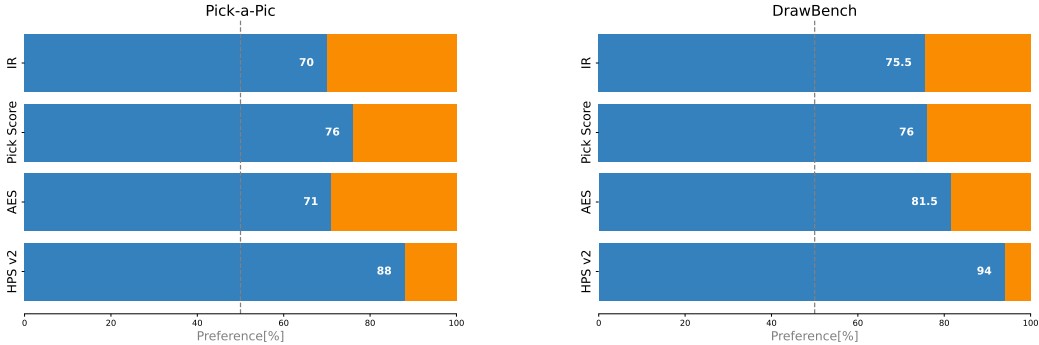

Figure 6: The winning rates of Z-Sampling over standard sampling. The blue bars represent the side of our method. The orange bars represent the side of the standard sampling. Model: DreamShaper-xl-v2-turbo. We present more results in Appendix D.3

Table 1: The quantitative results of Z-Sampling on Pick-a-Pic and DrawBench.

| Method | | Pick-a-Pic | | | | DrawBench | | | |
|---|---|---|---|---|---|---|---|---|---|
| | | HPS v2 ↑ | AES ↑ | PickScore ↑ | IR ↑ | HPS v2 ↑ | AES ↑ | PickScore ↑ | IR ↑ |
| SD-2.1 | Standard | 23.05 | 5.28 | 19.08 | -43.66 | 23.90 | 5.20 | 20.49 | -44.34 |
| | Resampling | 24.46 | 5.46 | 19.51 | **-18.07** | 23.94 | 5.08 | 20.40 | -30.90 |
| | Z-Sampling(ours) | **24.53** | **5.47** | **19.51** | -18.62 | **24.67** | **5.29** | **20.82** | **-23.61** |
| SDXL | Standard | 29.89 | 6.09 | 21.63 | 58.65 | 28.81 | 5.56 | 22.31 | 60.75 |
| | Resampling | 30.54 | 6.04 | 21.73 | 78.60 | 29.62 | 5.58 | **22.52** | 72.69 |
| | Z-Sampling(ours) | **31.28** | **6.13** | **21.85** | **79.22** | **30.50** | **5.68** | 22.46 | **79.97** |
| DreamShaper -xl-v2-turbo | Standard | 30.04 | 5.93 | 21.59 | 66.18 | 26.85 | 5.28 | 21.77 | 40.22 |
| | Resampling | 31.42 | 6.04 | 21.95 | 82.43 | 28.55 | 5.39 | 22.32 | 64.69 |
| | Z-Sampling(ours) | **32.38** | **6.15** | **22.11** | **90.87** | **29.90** | **5.64** | **22.35** | **73.51** |
| Hunyuan-DiT | Standard | 30.82 | 6.20 | 21.88 | 94.22 | 30.22 | 5.70 | 22.29 | 82.63 |
| | Resampling | 31.10 | 6.19 | 21.87 | 95.51 | **30.72** | 5.68 | 22.32 | 95.82 |
| | Z-Sampling(ours) | **31.12** | **6.31** | **21.90** | **97.88** | 30.53 | **5.75** | **22.40** | **96.13** |

In Table 1, we evaluate our method against standard sampling and Resampling across various diffusion architectures, including U-Net, DiT, and distillation architectures. Z-Sampling achieves top

Table 2: The quantitative results of Z-Sampling on GenEval. Model: SDXL

| Method | Single object ↑ | Two object ↑ | Counting ↑ | Colors ↑ | Position ↑ | Color attribution ↑ | Overall ↑ |
|---|---|---|---|---|---|---|---|
| Standard | 97.50% | 69.70% | 33.75% | 86.71% | 10.00% | 18.00% | 52.52% |
| Resampling | 98.75% | **76.77%** | 38.75% | **88.30%** | 5.00% | 20.00% | 54.59% |
| Z-Sampling(ours) | **100.00%** | 74.75% | **46.25%** | 87.23% | **10.00%** | **24.00%** | **57.04%** |

Table 3: The quantitative results of Z-Sampling and Semantic-CFG. Model: SD-2.1. For fairness, we follow the default settings of Semantic-CFG with the 768×768 resolution and SD-2.1.

| Method | Pick-a-Pic | | | | DrawBench | | | |
|---|---|---|---|---|---|---|---|---|
| | HPS v2↑ | AES↑ | PickScore↑ | IR↑ | HPS v2↑ | AES↑ | PickScore↑ | IR↑ |
| Standard | 25.67 | 5.66 | 20.20 | 0.53 | 25.98 | 5.37 | 21.39 | 6.77 |
| Semantic-aware CFG | 26.02 | 5.65 | 20.28 | 2.03 | 26.03 | 5.37 | 21.38 | 9.39 |
| Z-Sampling(ours) | **27.05** | **5.74** | **20.41** | **36.89** | **26.71** | **5.45** | **21.55** | **25.42** |

Table 4: Z-Sampling can enhance the training-free AYS. Model: DreamShaper-xl-v2-turbo.

| Method | Pick-a-Pic | | | | DrawBench | | | |
|---|---|---|---|---|---|---|---|---|
| | HPS v2↑ | AES↑ | PickScore↑ | IR↑ | HPS v2↑ | AES↑ | PickScore↑ | IR↑ |
| Standard | 32.80 | 6.05 | 22.31 | 91.48 | 30.94 | 5.57 | 22.68 | 77.44 |
| Z-Sampling(ours) | **33.53** | **6.16** | **22.45** | **103.95** | **31.92** | **5.71** | **22.78** | **95.82** |
| AYS | 32.78 | 6.05 | 22.32 | 91.88 | 30.95 | 5.57 | 22.68 | 77.85 |
| AYS + Z-Sampling(ours) | **33.57** | **6.15** | **22.45** | **104.22** | **31.93** | **5.72** | **22.75** | **94.82** |

Table 5: Z-Sampling can enhance the training-based Diffusion-DPO. Model: SDXL.

| Method | Pick-a-Pic | | | | DrawBench | | | |
|---|---|---|---|---|---|---|---|---|
| | HPS v2↑ | AES↑ | PickScore↑ | IR↑ | HPS v2↑ | AES↑ | PickScore↑ | IR↑ |
| Standard | 29.89 | 6.09 | 21.63 | 58.65 | 28.81 | 5.56 | 22.31 | 60.75 |
| Z-Sampling(ours) | **31.28** | **6.13** | **21.85** | **78.22** | **30.50** | **5.67** | **22.46** | **79.97** |
| Diffusion-DPO | 31.41 | 5.60 | 22.00 | 90.28 | 29.80 | 5.66 | 22.47 | 85.94 |
| DPO + Z-Sampling(ours) | **31.60** | **6.08** | **22.18** | **94.48** | **30.35** | **5.67** | **22.47** | **93.34** |

performance across nearly all metrics and Figure 6 shows the winning rates across these two benchmarks, exceeding **88%** on HPS v2. Furthermore, for a more detailed comparison, we present results on GenEval (Ghosh et al., 2024), which serves as a challenging benchmark. As Table 2 show, Z-Sampling significantly enhances alignment in aspects such as counting, two-object relations, and color attribution, further demonstrating the effectiveness of our method.

We also compare our method with a recent sampling technique designed to enhance semantic injection. Shen et al. (2024) proposed Semantic-aware CFG, dividing the latent into independent semantic regions at each denoising step and adaptively adjusting their guidance, thereby unifying the effects across regions. While the setting is different from previous experiments, this results still underscore the effectiveness of Z-Sampling remains unaffected. As shown in Table 3, we observe that Z-Sampling demonstrates a higher improvement.

Moreover, we present more quantitative experimental results in Appendix D.1 and more qualitative comparison across various dimensions (e.g, color, style, and etc.) in Appendix D.2.

Specifically, we also discuss the effect of Z-Sampling under extremely high CFG guidance in Appendix D.4, demonstrating its ability to achieve a favorable balance between image quality and prompt adherence, suppressing artifacts and oversaturation.

**Orthogonal Methods** Z-Sampling can be combined with other orthogonal methods to further enhance diffusion models. In Table 4, Z-Sampling further enhances AYS-Sampling, a sampling strategy that optimizes the denoising scheduler, leading to improved overall performance. Note that AYS-Sampling only released the 10-step scheduler, which is more applicable to DreamShaper-

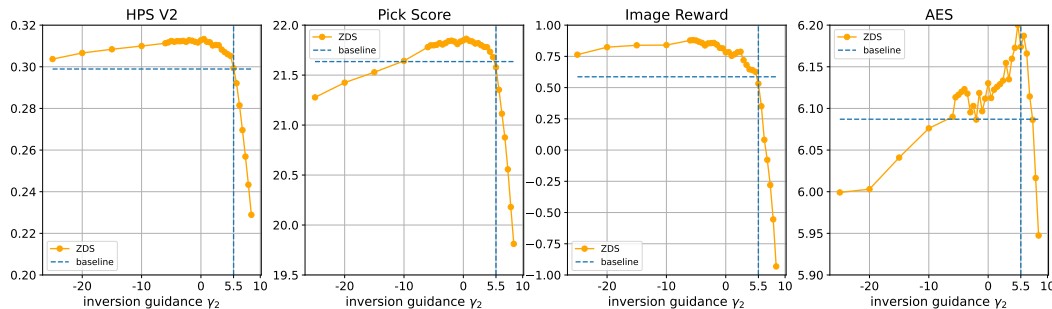

Figure 7: Robustness to the inversion guidance scale. When the gap is zero, i.e., the inversion guidance equals the denoising guidance (e.g. $\gamma_1 = \gamma_2$), the positive gains almost disappear.

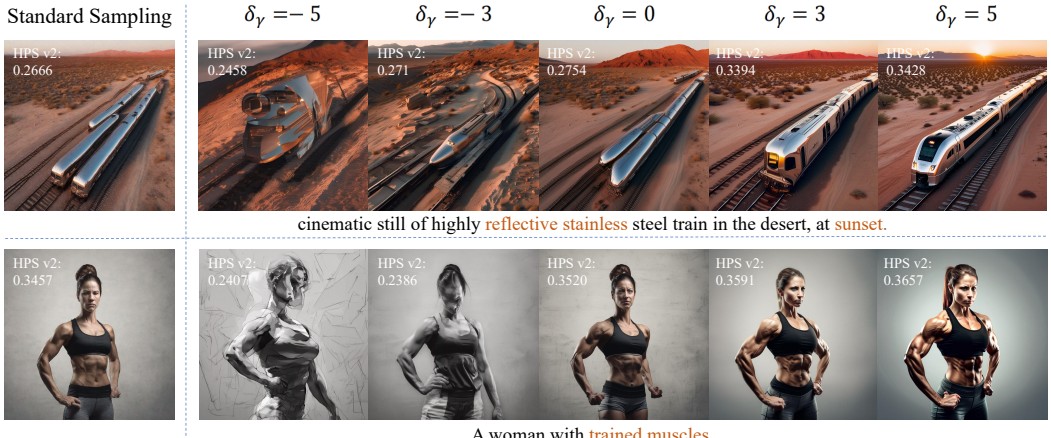

Figure 8: The guidance gap $\delta_\gamma$ between $\gamma_1$ and $\gamma_2$ influences both the magnitude and direction of semantic injection. When $\delta_\gamma$ is large ($\delta_\gamma$=5), the gain of Z-Sampling becomes pronounced. Conversely, when $\delta_\gamma$ is zero or even negative, it approximately degenerates into standard sampling or significantly break generation.

v2-turbo. Additionally, Table 5 shows that Z-Sampling can also be combined with training-based methods, further enhancing the generation quality of Diffusion-DPO. We leave more quantitative results of enhancing orthogonal methods in Table 8.

**The Guidance Gap** We first examine the impact of guidance scale. In Section 3.1, we show that the guidance gap between denoising and inversion dictates the degree of semantic information gain. To further verify this, we fix the guidance scale $\gamma_1$ as 5.5 following standard sampling. By varying $\gamma_2$, we control the guidance gap $\delta_\gamma = \gamma_1 - \gamma_2$ to observe its impact. As shown in Figure 7, when $\gamma_2$ increases and the guidance gap $\delta_\gamma$ narrows, the benefits of Z-Sampling diminish. According to the theoretical results of semantic information gain, a zero guidance gap can approximately lead to standard sampling. When the gap is below zero ($\gamma_2 > \gamma_1$), it can result in a negative gain. In Figure 8, we present a qualitative analysis showing that when the zero guidance gap indeed yields very similar results to standard sampling.

**Zigzag Diffusion Steps** We note that $\lambda$ indicates the first $\lambda$ steps using the zigzag operation. For example, when $\lambda$ is 0, it reverts to standard SDXL. When $\lambda$ is 25, it means the first 25 steps of the denoising process use the zigzag operation. We conducted experiments on Pick-a-Pick using SDXL (50 steps), as shown in Figure 9, when $\lambda$ increases from 0 to 25, the winning rate rises from 50% to 75%. However, when $\lambda$ increases from 25 to 50 steps, it only rises from 75% to 80%. This indicates that the zigzag operation is more effective during the early stages of denoising process.

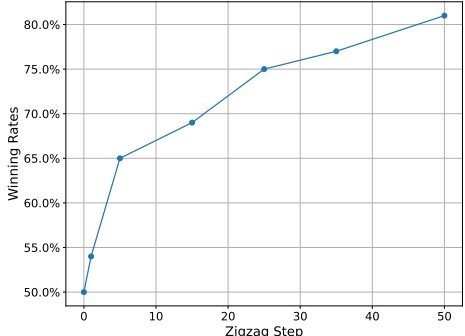

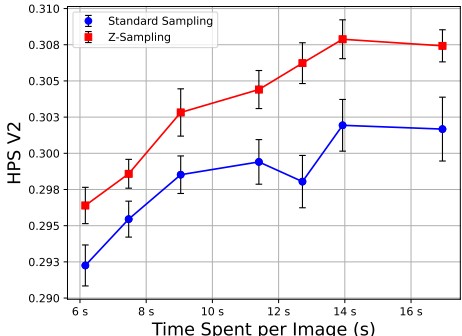

Figure 9: Robustness to the zigzag diffusion steps $\lambda$. The horizontal axis shows the number of zigzag operations, and the vertical axis represents the winning rate over HPS v2 on Pick-a-Pic. As $\lambda$ increases, generation quality improves, indicating effective semantic information gain throughout the whole path.

Figure 10: Z-Sampling outperforms standard sampling with the same time consumption and significantly enhance the performance peak of pretrained SDXL. The horizontal axis shows the average time per image with various denoising steps, while the vertical axis shows the average HPS v2 on the Pick-a-Pic benchmark.

**Efficiency Comparison** When the denoising steps are fixed (e.g., $T$=50), Z-Sampling naturally incurs additional time consumption due to the zigzag step. Suppose the timestep lengths for Standard Sampling and Z-Sampling are $T_s$ and $T_z$, respectively, then the corresponding noise prediction operation times are $T_s$ and $T_z + 2\lambda$. To facilitate a fairer comparison in terms of computation time, we set $T_z = \frac{1}{2}T_s$ and $\lambda = \frac{1}{2}T_z$. This allows us to compare evaluation scores under the same generation time consumption per image. Figure 10, indicates that Z-Sampling significantly outperforms standard sampling and enhance the performance peak. Even with 36% less computational time, Z-Sampling can surpass the best performance of standard sampling.

## 5 DISCUSSION AND CONCLUSION

**Discussion** We further discuss the limitations and future directions of our work. First, we note that Z-Sampling relies on the semantic information gain through deterministic inversion, limiting its applicability to deterministic samplers, such as DDIM. Extending it to the SDE-based diffusion framework is an important direction for future work (see Appendix E.1). Second, while Z-Sampling exhibits strong generalization, we only studied text-to-image diffusion models in this work. Therefore, exploring its applications to areas such as video generation, 3D generation, and molecular synthesis is naturally another promising research direction. However, due to the different natures of latent space and sampling schedulers, this direction may require further algorithm design and theoretical understanding. Third, while the extra computational cost is acceptable according to the experiments, Z-Sampling takes more computational time on its zigzag steps. It will be interesting to distill the path of Z-Sampling into the model itself.

**Conclusion** To the best of our knowledge, this work is the first to theoretically and empirically discover that the guidance gap between denoising and inversion can inject semantic information into the latent space, which can lead to improved generation. By theoretically investigating how the semantic information gain depends on the guidance gap, we naturally derive Z-Sampling, a novel self-reflection-based diffusion sampling method that can accumulate semantic information through zigzag self-reflection operation and, thus, generate more desirable results. The extensive experiments not only demonstrate that various models can self-improve significantly with Z-Sampling in various settings, but also suggest that Z-Sampling can further enhance other orthogonal methods. In summary, Z-Sampling is flexible, additive, and powerful with limited coding an computation costs. Given the theoretical mechanism and empirical success of Z-Sampling and diffusion self-reflection, we believe this work can motivate better theoretical understanding of visual generation and inspire more advanced sampling methods. Moreover, this approach will soon incentivize video generation, 3D generation, and beyond.

ETHICS STATEMENT

We propose Z-Sampling, a novel guidance mechanism designed to enhance the quality of diffusion model generation. Although it does not directly involve human subjects or issues related to dataset privacy, we have carefully considered its potential ethical and moral implications. We ensure the transparency of all datasets used for debugging and developing the algorithm, and their randomness guarantees the absence of bias in the ethical domain, which is of utmost importance. Additionally, all models used comply with the terms of open-source licenses. Given Z-Sampling's significant commercial potential, we strive to apply this technology responsibly, ensuring that its applications yield positive societal benefits.

ACKNOWLEDGEMENT

This work was supported by the Science and Technology Bureau of Nansha District Under Key Field Science and Technology Plan Program No. 2024ZD002. This work was supported by Guangdong Provincial Key Lab of Integrated Communication, Sensing and Computation for Ubiquitous Internet of Things(No.2023B1212010007).

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

## A   EXPERIMENTAL DETAILS

In this section, we introduce the details of the metrics and benchmarks used in the experiments.

### A.1   DATASETS

**Pick-a-Pic.**   The Pick-a-Pic dataset (Kirstain et al., 2023) was generated by logging user interactions with the Pick-a-Pic web application for text-to-image generation. Each entry includes a prompt, two generated images, and a label indicating the preferred image or a tie if neither is significantly favored. Here we use only the first 100 prompts as the test set, which is sufficient to reflect the model's capabilities.

**Drawbench.**   DrawBench is a comprehensive and challenging benchmark for text-to-image models, introduced by the Imagen research team (Saharia et al., 2022). It contains 11 categories, including aspects such as color, counting, and text, with approximately 200 text prompts.

**GenEval.** Geneval (Ghosh et al., 2024) is an object-focused framework designed to evaluate compositional properties of images, including object co-occurrence, position, count, and color. It incorporates 553 prompts, achieving an 83% agreement with human judgments regarding the correctness of the generated images[*].

**PartiPrompts.** PartiPrompts (Yu et al.) is a collection of over 1,600 diverse prompts in English, designed to assess the capabilities of models across different categories and challenges. The prompts cover a wide range of topics and styles, helping evaluate the strengths and weaknesses of models in areas like language understanding, creativity, coherence. Here we randomly select 100 prompts from Part for evaluation.

## A.2 METRICS

**AES.** Aesthetic score (AES) (Schuhmann et al., 2022) refers to a mechanism for evaluating the visual quality of generated images, which assigns a quantitative score based on attributes like contrast, composition, color, and detail, reflecting alignment with human aesthetic standards.

**PickScore.** Kirstain et al. (2023) developed Pick-a-Pic, a large open dataset consisting of text-to-image prompts and real user preferences for generated images. They then utilized this dataset to train a CLIP-based scoring function, PickScore, for the task of predicting human preferences.

**ImageReward.** Xu et al. (2024a) developed ImageReward, the first general-purpose text-to-image human preference reward model. which is trained based on systematic annotation pipeline, including rating and ranking and has collected 137,000 expert comparisons to date.

**HPS v2.** Wu et al. (2023c) first introduced the Human Preference Dataset v2 (HPD v2), a large-scale dataset comprising 798,090 human preference choices on 433,760 pairs of images. By fine-tuning CLIP using HPD v2, they developed the Human Preference Score v2 (HPS v2), a scoring model that more accurately predicts human preferences for generated images.

## A.3 BASELINES

**Semantic-aware CFG** (Shen et al., 2024), adaptively adjust the CFG scales across different semantic regions to mitigate the undesired effects caused by guidance.

**Diffusion-DPO** (Wallace et al., 2024), finetune a pretrained Diffusion model using carefully curated high quality images and captions to improve visual appeal and text alignment.

**AYS-Sampling** (Sabour et al., 2024), a strategy for optimizing sampler timesteps, which accounts for the dataset, model, and sampler to enhance image quality.

**Semantic-Aware CFG** (Shen et al., 2024), a strategy dividing the latent into independent semantic regions at each denoising step and adaptively adjusting their guidance, thereby unifying the effects across regions.

**Smoothed Energy Guidance** (Hong, 2024), which employs energy landscape perspective and intermediate self-attention maps to achieve higher quality samples.

**CFG++** (Chung et al., 2024), which optimizes the classifier-free guidance mechanism from the perspective of manifold constraints. It only replaces the conditional latent with the unconditional latent during the classifier free guidance mechanism, effectively addressing the oversaturation issue caused by high CFG values.

---

[*]To ensure consistency with other experiments, we used a denoising guidance scale of 5.5, differing from the default 9.0 in GenEval.

## B    RELATED WORKS

In this section, we discuss existing work related to Z-Sampling.

**Semantic Information in Latent Space**    Recent works have shown that the prior information present in the noise latent can significantly impact the quality of image generation (Xu et al., 2024b; Mao et al., 2023a; Samuel et al., 2024; Zhou et al., 2024; Liu et al., 2024a). For example, Mao et al. (2023b) found certain regions in random latents can induce objects representing specific concepts. And Zhou et al. (2024) propose Golden Noise by leveraging a semantic accumulation approach. And (Po-Yuan et al., 2023) found slight perturbations can lead to significant changes in the diffusion model's generated results. And injecting semantic information (e.g., low-frequency wavelengths) into Gaussian noise can enhance image quality, particularly improving alignment performance (Wu et al., 2023c; Guttenberg, 2023; Lin et al., 2024a; Qi et al., 2024). IRFDS (Yang et al., 2024) utilizes a pretrained rectified flow model to provide a prior, optimizing the initial latent for image editing task. Building on these studies, we investigate semantic information from the guidance perspective, implicitly integrating it into the generation process without requiring explicit reference data.

**Sampling Strategies of Diffusion Model**    To improve the sampling process, lug (2022) proposed Resampling that involves adding random noise and performing multiple back-and-forth samples at each timestep. Subsequent studies adopted this paradigm for tasks such as video generation (Wu et al., 2023b) and universal classifier guidance (Bansal et al., 2023). IRFDS (Yang et al., 2024) utilizes a pretrained rectifying flow model to provide a prior, optimizing the initial latent for better image editing. However, they overlooked the importance of inverted latent and simply applied random noise, which does not effectively enhance prompt adherence. In Tune-a-Video, to ensure structural consistency, Wu et al. (2023a) incorporate the denoising-inversion paradigm as a subcomponent. However, their end-to-end approach is not optimal and overlooks the importance of the guidance gap. To reduce spatial inconsistency in different latent regions under the same guidance scale, Shen et al. (2024) developed adaptive guidance based on semantic segmentation. It relies on attention-level changes, limiting adaptability to other algorithms, and its robustness is influenced by semantic segmentation effectiveness. Constraint-based approaches aim to improve sampling, for example, Chung et al. (2024) substitutes conditional noise with unconditional noise to enhance generation quality from an image manifold perspective, though improvements are minimal. Yang et al. applies spherical gaussian constraint during guidance, but it requires a reference data, limiting its applicability. Finally, we note that Z-Sampling can be effectively transferred to other generative paradigms, such as Masked Generative Models (Shao et al., 2024b) and IV-Mixed Sampler (Shao et al., 2024a).

## C    MOTIVATION AND OBSERVATION

### C.1    LATENTS WITH SEMANTIC INFORMATION

In Figure 11, we present additional cases illustrating that random latents encode relevant semantic information. For instance, for prompts related to the concept "Jeep Cars", the latent corresponding to seed 20 achieves the highest performance, with PickScore of 23.4784, whereas latents from other seeds fail to exceed PickScore of 23.

### C.2    INVERSION MAKES GOOD LATENTS

In this section, we show that the inverted latent inherently carries semantic information related to the conditional prompt $c$. These extra semantic information gain leads to superior generation outcomes.

First, we choose images of "cats" and "spiders" as depicted in Figure 12. Employing the DDIM inversion algorithm with guidance scale set to 0, we obtain $latent_{inv\_1}$ and $latent_{inv\_2}$. We hypothesize that $latent_{inv\_1}$ encapsulates semantic information associated with "cat" whereas $latent_{inv\_2}$ inherently relates more closely to "spiders".

Next, we use these two latents to generate images conditioned on text prompts "cats" and "spiders" respectively, as illustrated in Figure 13. We observe that $latent_{inv\_1}$ performs better when condi-

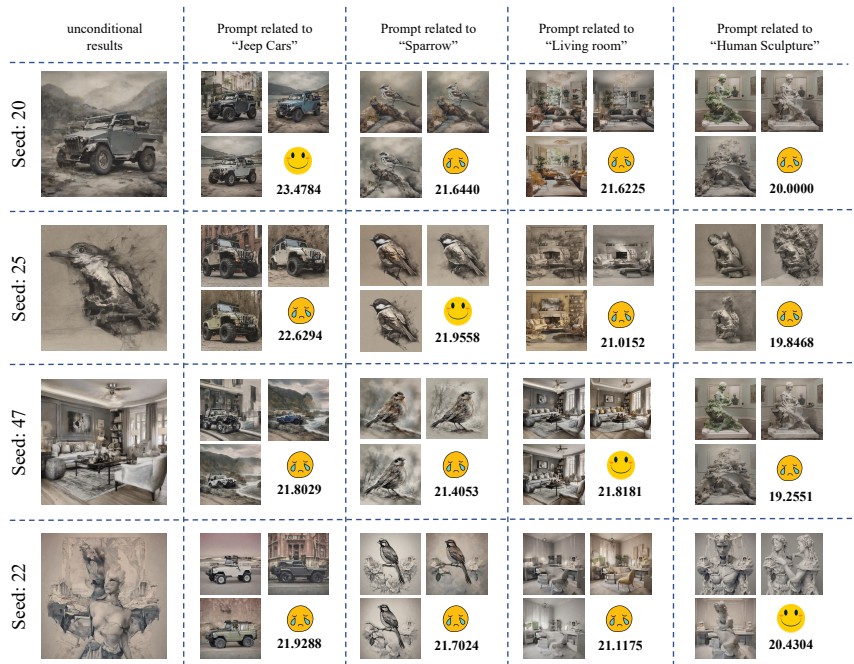

Figure 11: Latents with relevant semantic information about a specific concept can generate images more effectively from prompts related to that concept. Each row shows the results of the same latent across different prompts, while each column shows results from different latents under the same prompts. For each cell, we compute the PickScore. For example, the latent related seed 20 achieves an PickScore of 23.4784 when generating images related to "Jeep Cars".

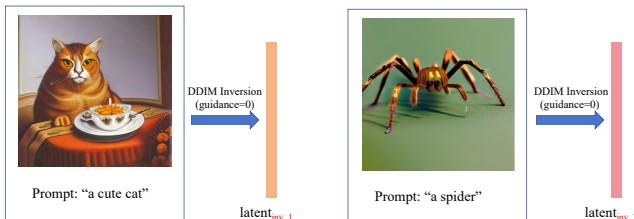

Figure 12: Given two natural images and their corresponding prompts, we perform DDIM inversion to reverse them and obtain the corresponding initial noise latents.

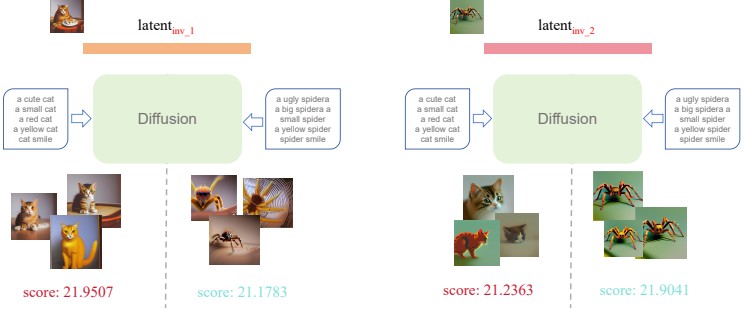

Figure 13: Generate images related to "cat" and "spider" using two latents respectively, and calculate the PickScore.

tioned on text related to "cats" while latent$_{inv\_2}$ performs better when conditioned on text related to "spiders". This phenomenon empirically validates our hypothesis that inverted latent does matter.

## D  SUPPLEMENTARY EXPERIMENTAL RESULTS

In this section, we present more quantitative and qualitative results of Z-Sampling.

### D.1  SUPPLEMENTARY QUANTITATIVE RESULTS

**Results of Z-Sampling in other benchmarks**   In Table 6, we evaluate 100 randomly selected prompts from PartiPrompts using the SDXL model, with Z-Sampling demonstrating the higher performance. Additionally, we also compare classical metrics such as FID (Seitzer, 2020), IS (Salimans et al., 2016), and clip-score (Radford et al., 2021) on MS-COCO 2014 (Lin et al., 2014). Due to numerous evaluation prompts (30K), we employ the distilled model, DreamShaper-xl-v2-turbo, with 4 denoising steps, showing the higher generation quality in Table 7. We also report additional comparative results on Geneval in Table 8, including Resampling and Diffusion-DPO, showcasing Z-Sampling's superiority in average scores.

Table 6: The quantitative results of Z-Sampling on PartiPrompts. Model: SDXL.

| Method | HPS v2 ↑ | AES ↑ | PickScore ↑ | IR ↑ |
|---|---|---|---|---|
| Standard | 29.34 | 5.81 | 22.27 | 72.53 |
| Resampling | 30.21 | 5.78 | 22.42 | 92.34 |
| Z-Sampling(ours) | **31.00** | **5.85** | **22.43** | **97.32** |

Table 7: The quantitative results of Z-Sampling on MS-COCO 2014. Model: DreamShaper-xl-v2-turbo.

| Method | IS-30K ↑ | FID-30K ↓ | Clip-Score ↑ |
|---|---|---|---|
| Standard | 34.0745 | 24.1420 | 0.3267 |
| Z-Sampling(ours) | **34.4173** | **23.4958** | **0.3288** |

Table 8: The additional quantitative results of Z-Sampling on GenEval. Model: SDXL

| Method | Single object ↑ | Two object ↑ | Counting ↑ | Colors ↑ | Position ↑ | Color attribution ↑ | Overall ↑ |
|---|---|---|---|---|---|---|---|
| Standard | 97.50% | 69.70% | 33.75% | 86.71% | 10.00% | 18.00% | 52.52% |
| Diffusion-DPO | 100.00% | 80.81% | 45.00% | 88.30% | 10.00% | **31.00%** | 59.18% |
| DPO+Z-Sampling(ours) | **100.00%** | **82.83%** | **46.25%** | **89.36%** | **10.00%** | 29.00% | **59.57%** |

**Results of Z-Sampling in other baselines and tasks**   We also compare Z-Sampling with other methods that improve the effect of guidance. Specifically, Hong et al. (2022) proposed SAG, which employs blur guidance and intermediate self-attention maps to achieve higher quality samples. Furthermore, SEG (Hong, 2024) further optimized SAG from the energy landscape perspective. Here we report the comparison results with SEG in Table 9. Additionally, We have also compared Z-Sampling with CFG++ (Chung et al., 2024), which optimizes the classifier-free guidance mechanism from the perspective of manifold constraints. since it restricts the cfg scale to the range from 0.0 to 1.0, while the classic Z-Sampling is larger, a fair comparison is not possible. Given this, we use $\omega = 0.5$ in CFG++, corresponding to a cfg scale of 5.5 in Z-Sampling.

Table 9: The quantitative results of Z-Sampling and SEG. Model: SDXL.

| Method | Pick-a-Pic | | | | DrawBench | | | |
|---|---|---|---|---|---|---|---|---|
| | HPS v2↑ | AES↑ | PickScore↑ | IR↑ | HPS v2↑ | AES↑ | PickScore↑ | IR↑ |
| Standard | 29.89 | 6.09 | 21.63 | 58.65 | 28.81 | 5.55 | 22.31 | 60.75 |
| SEG | 30.53 | 6.12 | 21.42 | 61.57 | 29.60 | 5.66 | 22.15 | 60.42 |
| Z-Sampling(ours) | **31.28** | **6.13** | **21.85** | **78.22** | **30.50** | **5.67** | **22.46** | **79.97** |

Finally, as a general method, we test Z-Sampling's performance on the video generation task. We choose AnimateDiff (Guo et al., 2023) as the baseline model and test it on Chronomagic-Bench-150 (Yuan et al., 2024), and we set $\gamma_1 = 7.5$ and $\gamma_2 = 0$ in Z-Sampling. With the results shown in Table 11, we note that Z-Sampling outperforms both AnimateDiff and another train-free sampling method FreeInit (Wu et al., 2025) in UMT-FVD (Liu et al., 2024b), UMT-SCORE (Li et al., 2023), GPT4o-MTSCORE (Achiam et al., 2023).

Table 10: The quantitative results of Z-Sampling and CFG++. Model: SDXL. It is worth noting that in the official implementation of CFG++, the VAE encoder uses **madebyollin/sdxl-vae-fp16-fix** checkpoint. For fair comparison, we follow this setting, so the results reported for SDXL and Z-Sampling are slightly different from the previous results.

| Method | Pick-a-Pic | | | | DrawBench | | | |
|---|---|---|---|---|---|---|---|---|
| | HPS v2↑ | AES↑ | PickScore↑ | IR↑ | HPS v2↑ | AES↑ | PickScore↑ | IR↑ |
| Standard | 30.04 | 6.11 | 21.80 | 60.07 | 28.85 | 5.62 | 22.42 | 67.61 |
| CFG++ | 30.28 | 6.09 | 21.83 | 67.30 | 28.65 | 5.62 | 22.38 | 62.66 |
| Z-Sampling(ours) | **31.24** | **6.12** | **21.85** | **78.55** | **30.35** | **5.66** | **22.44** | **79.11** |

Table 11: The quantitative results of Z-Sampling on Chronomatic-Bench-150. Model: AnimateDiff.

| Method | UMT-FVD ↓ | UMT-SCORE ↑ | GPT4o-MTSCORE ↑ |
|---|---|---|---|
| Standard | 275.18 | 2.82 | 2.83 |
| FREEINIT | 268.31 | 2.82 | 2.59 |
| Z-Sampling(ours) | **243.26** | **2.97** | **2.88** |

Table 12: The quantitative results of Z-Sampling under different backtracking stepsize k. Model: SDXl.

| k | HPS v2 ↑ | AES ↓ | PickScore ↑ | IR ↑ |
|---|---|---|---|---|
| 0 (SDXL) | 29.89 | 6.09 | 21.64 | 58.65 |
| 1 | **31.28** | **6.13** | **21.85** | 79.22 |
| 2 | 31.11 | 6.08 | 21.72 | **84.53** |
| 3 | 30.75 | 6.09 | 21.48 | 78.54 |
| 4 | 30.59 | 6.09 | 21.34 | 78.60 |

**Multiple steps of denoising and inversion operation in Z-Sampling** We have explored the one-step scenario, i.e, $x_t \rightarrow x_{t-1} \rightarrow \tilde{x}_t$. Here, we extend to multiple steps scenario, i.e., $x_t \rightarrow x_{t-k} \rightarrow \tilde{x}_t$. As shown in Table 12, the best performance is achieved when k=1. As k increases, the performance of Z-Sampling deteriorates, which aligns with the Theorem 1 and Theorem 2, where increasing k gradually brings the step-by-step approach closer to end-to-end, thereby increasing the error term $\tau_2$. Specifically, when k=T-1 and the zigzag operation is only performed on the initial latent, it corresponds to the scenario in Table 17.

## D.2 Supplementary Qualitative Results

In Figure 14, we note Z-Sampling can better recognize the stylistic descriptions in prompts. For example, it can generate "Mario characters" that are more realistic and lifelike.

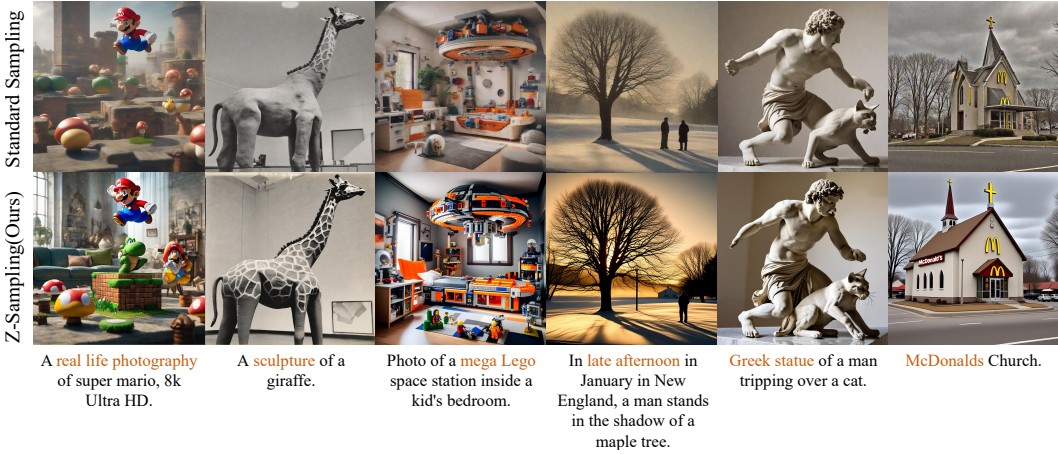

A real life photography of super mario, 8k Ultra HD. | A sculpture of a giraffe. | Photo of a mega Lego space station inside a kid's bedroom. | In late afternoon in January in New England, a man stands in the shadow of a maple tree. | Greek statue of a man tripping over a cat. | McDonalds Church.

Figure 14: Qualitative comparison in terms of style.

In Figure 15, we note Z-Sampling accurately interprets object positional relationships, e.g., 'underneath', 'on top of', 'on the right of', etc.

In Figure 16, Z-Sampling enhances the binding of color attributes, aligning images more closely with prompts and improving quality.

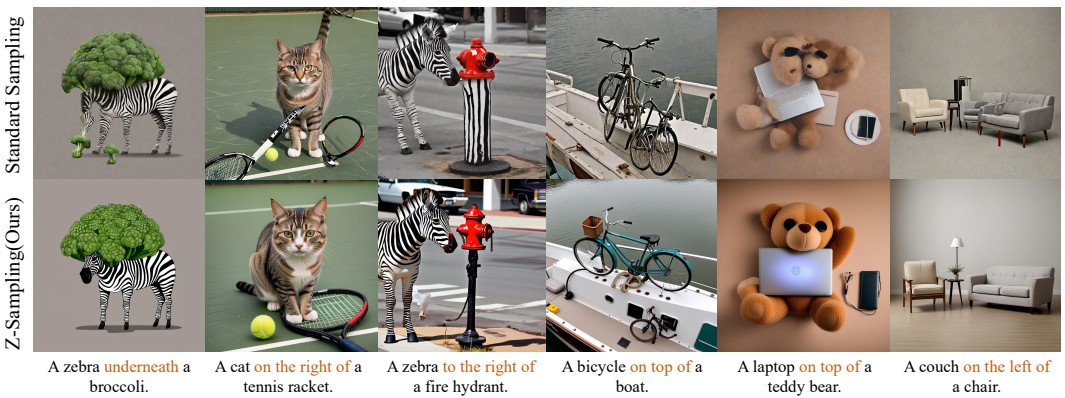

Figure 15: Qualitative comparison in terms of position.

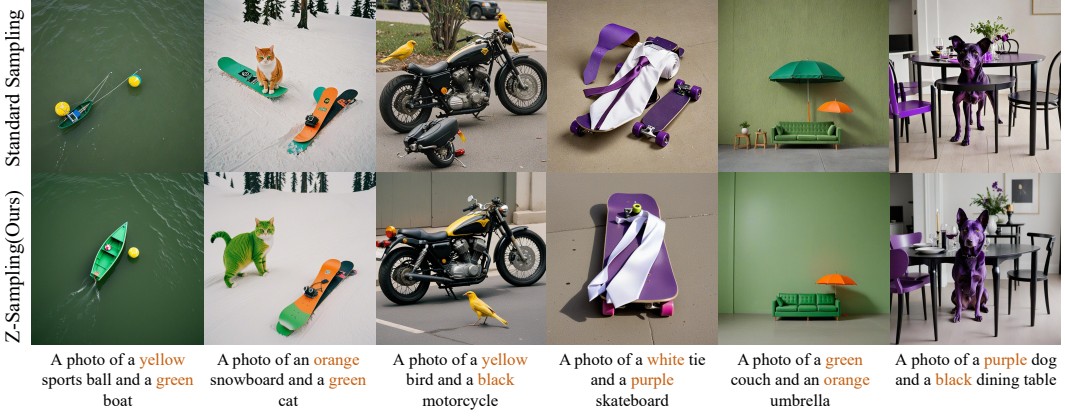

Figure 16: Qualitative comparison in terms of color.

In Figure 17, we note Z-Sampling demonstrates enhanced capability in understanding quantitative relationships, effectively addressing the persistent challenge in diffusion models. For example, it can effectively understand and generate images such as 'three suitcases', 'four buses', and two beds'.

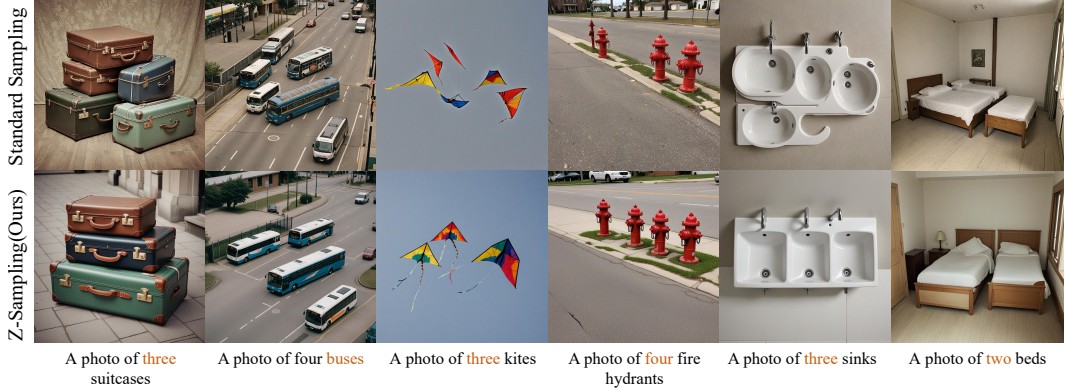

Figure 17: qualitative comparison in terms of counting.

In Figure 18, we find that Z-Sampling aids in generating Multi-object composite (e.g., a mouse and a bowl) or counterfactual (e.g., an elephant in the sea) images, manifested in its enhanced 'co-occurrence' capability.

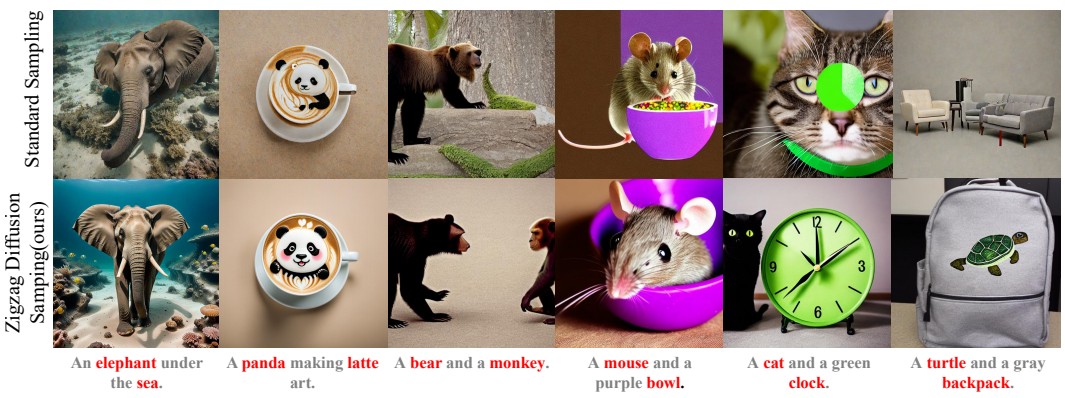

Figure 18: Qualitative comparison in terms of object co-occurrence.

### D.3 WINNING RATES COMPARISON

Here, we present a comparative analysis of winning rates under various settings, such as different models and denoising steps. The blue bars represent Z-Sampling (ours), while the orange bars represent the standard sampling method. Winning rates of our method exceeds 50% in all metrics. Especially HPS v2, which is much better than standard method.

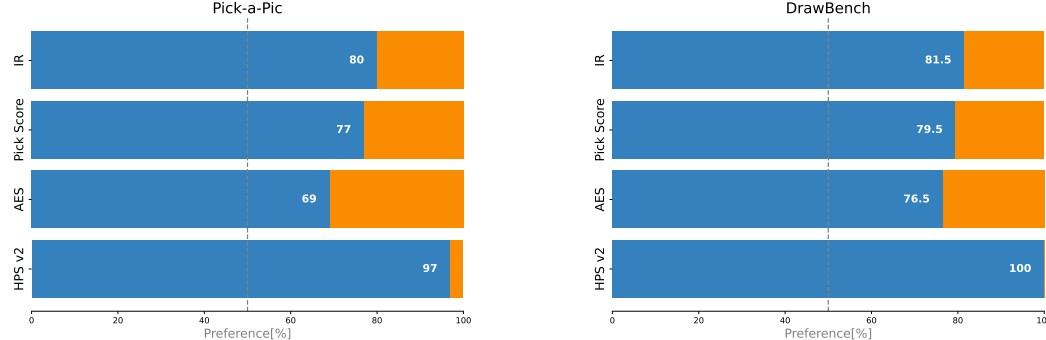

Figure 19: Comparison of Winning Rates with 10 Denoising Steps in the SDXL.

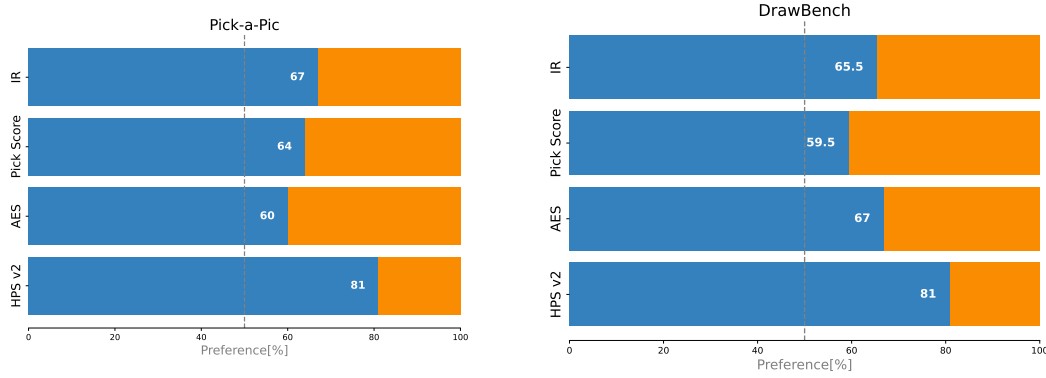

Figure 20: Comparison of Winning Rates with 50 Denoising Steps in the SDXL.

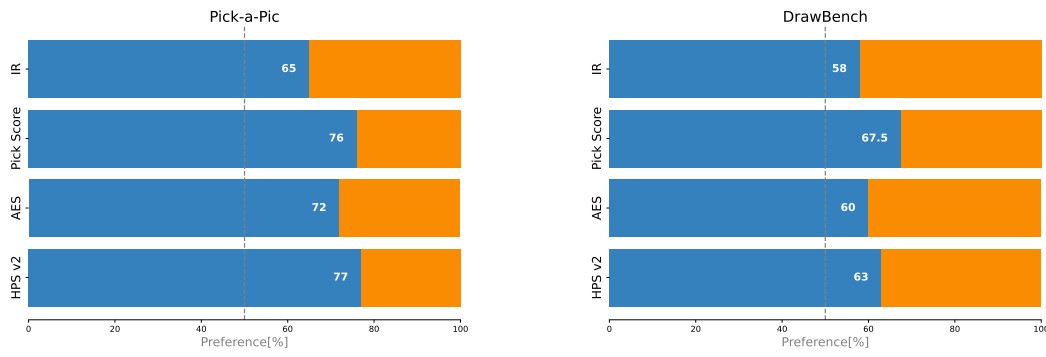

Figure 21: Comparison of Winning Rates with 50 Denoising Steps in the SD 2.1.

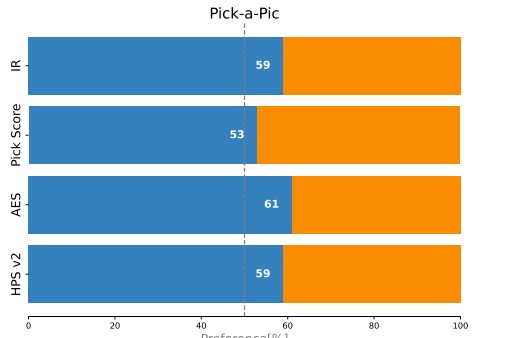 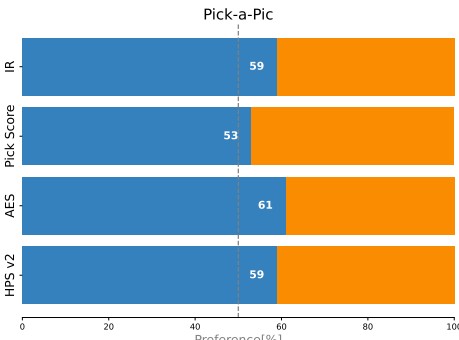

Figure 22: Comparison of Winning Rates with 10 Denoising Steps in the Hunyuan-DiT.

### D.4 PERFORMANCE OF Z-SAMPLING UNDER HIGH CFG SCALE

We also report the performance of Z-Sampling under different intensities of classifier free guidance $\gamma_1$ during denoising process.

We use DreamShaper-xl-turbo-v2 as the base model. As shown in Table 13, the standard sampling performs best at $\gamma_1 = 3.5$, which is also the official recommended guidance sclae. When $\gamma_1 \geq 3.5$, the standard sampling begins to exhibit issues such as oversaturation and artifacts.

However, Z-Sampling consistently yields positive gains, indicating that our method can still work effectively under high guidance scales. And we present the winning rate of Z-Sampling over Standard sampling on HPS v2 across different guidance sclae $\gamma_1$ in Figure 23, further validating this point.

Table 13: Performance of Z-Sampling under different guidance $\gamma_1$. Model: DreamShaper-xl-turbo-v2. We note that the official recommended guidance scale $\gamma_1 = 3.5$. When $\gamma_1 > 3.5$, the quality of standard sampling gradually declines, while Z-Sampling still shows improvement on this basis.

| Method | $\gamma_1$ | HPS v2 ↑ | AES ↑ | PickScore ↑ | IR ↑ | Winning Rate↑ |
|---|---|---|---|---|---|---|
| Standard Sampling | 1.5 | 28.51 | 5.83 | 21.37 | 43.25 | - |
| Z-Sampling | 1.5 | **29.51** | **6.02** | **21.66** | **55.89** | 73% |
| Standard Sampling | 3.5 | 30.04 | 5.94 | 21.59 | 66.18 | - |
| Z-Sampling | 3.5 | **32.38** | **6.15** | **22.11** | **90.87** | 88% |
| Standard Sampling | 5.5 | 29.96 | 5.97 | 21.37 | 64.46 | - |
| Z-Sampling | 5.5 | **31.42** | **6.05** | **21.83** | **76.00** | 85% |
| Standard Sampling | 7.5 | 29.10 | 5.88 | 21.02 | 60.26 | - |
| Z-Sampling | 7.5 | **30.90** | **5.96** | **21.59** | **74.18** | 86% |
| Standard Sampling | 9.5 | 27.98 | 5.76 | 20.59 | 41.70 | - |
| Z-Sampling | 9.5 | **29.95** | **5.88** | **21.28** | **63.40** | 92% |
| Standard Sampling | 11.5 | 26.93 | 5.60 | 20.30 | 31.45 | - |
| Z-Sampling | 11.5 | **28.97** | **5.77** | **20.97** | **55.69** | 91% |

Generally, classifier-free guidance serves as a mechanism for semantic control, balancing image quality and prompt adherence, with excessive guidance scale causing deviations and artifacts. Z-Sampling, as a similar semantic enhanced mechanism, employs an iterative approach (unlike the vanilla CFG mechanism, which directly alters the latent distribution) to more effectively explore this balance. And we presents some visual cases in Figure 24, showcasing Z-Sampling's capability to maintain image quality even under high guidance scale.

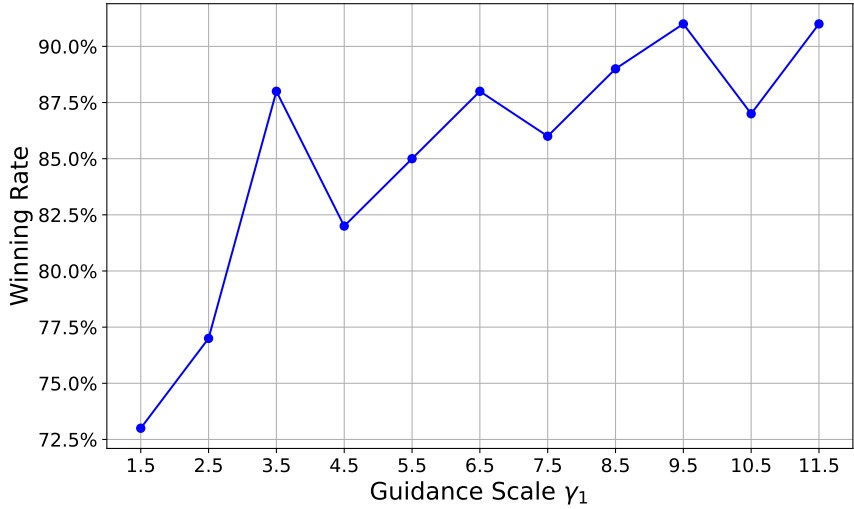

Figure 23: Comparison of Winning Rates under different guidance scale $\gamma_1$. Model: DreamShaper-xl-turbo-v2. Horizontal axis: guidance scales $\gamma_1$. Vertical axis: Z-Sampling vs Standard Sampling winning rates on Pick-a-Pic.

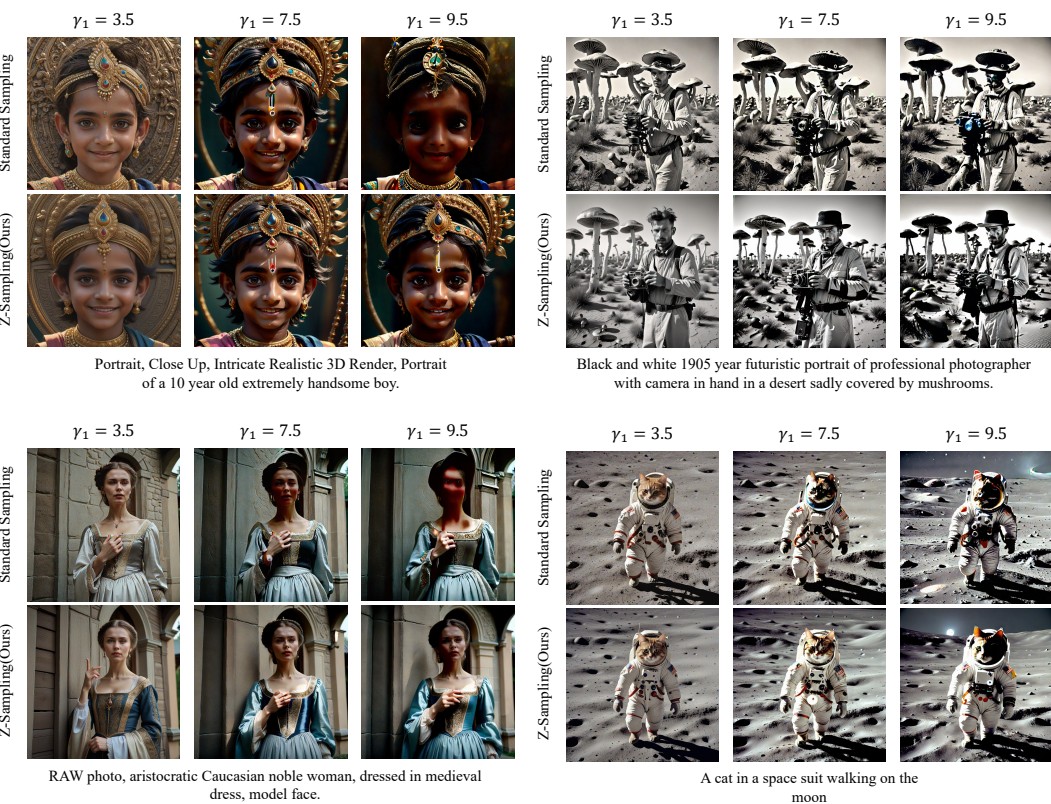

Figure 24: Qualitative comparison under high guidance scale. When $\gamma_1 = 3.5$ (the official recommended guidance scale), both Z-Sampling and Standard exhibit no artifacts or degradation in image quality. As $\gamma_1$ increases, standard sampling exhibits artifacts and oversaturation, while Z-Sampling is less affected.

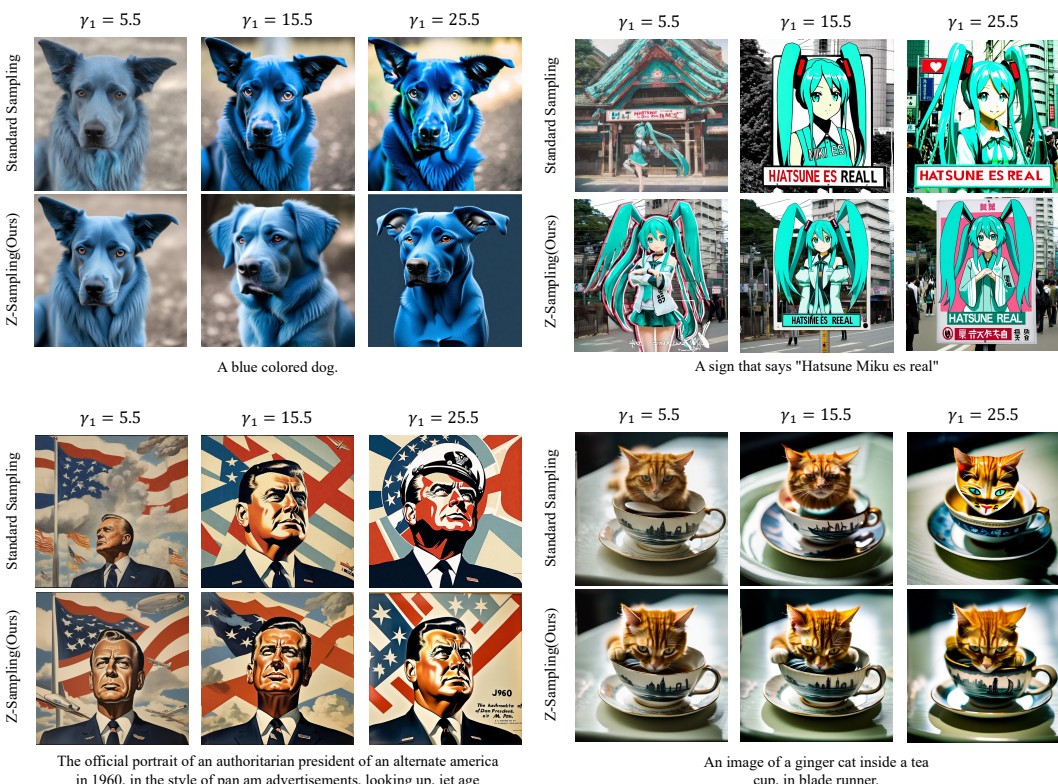

Figure 25: Qualitative comparison under high guidance scales. Standard sampling suffers from supersaturation more significantly than Z-Sampling under the high guidance scales. Model:SDXL.

## D.5 ADDITIONAL EXPERIMENTS ON VARIOUS GUIDANCE SCALES

We report more visual cases in Figure 25, showcasing the performance of Z-Sampling in SDXL under different guidance scales $\gamma_1$. It can be observed that as the guidance scale increases, the phenomenon of artifacts and oversaturation for standard sampling become more pronounced, while Z-Sampling effectively mitigates these issues. The similar observation also holds in Figure 24 with DreamShaper.

To further investigate the performance improvement with various high CFG scales, we present the conclusive quantitative experimental results of SD 2.1, SDXL, and DreamShaper together in the additional Table 14. We searched the best guidance scales for each model in terms of HPS v2 and present the results. For SDXL/SD2.1, the seached guidance range was set from 3.5 to 25.5, and for DreamShaper-xl-turbo-v2, it was set from 1.5 to 11.5. Note that existing relevant studies commonly do not fine-tune the guidance scale hyperparameter.

The conclusive quantitative results demonstrate that Z-Sampling can significant improve the best performance of all three diffusion models with various choices of the guidance scales. Moreover, the results indicate that the distilled DreamShaper with Z-Sampling can even outperform SDXL, while DreamShaper with standard sampling cannot match SDXL.

## E ANALYSIS OF THE APPROXIMATION ERROR TERM

In this section, we undertake a more in-depth analysis of the approximation error term $\tau_2$ within Equation 10. We first demonstrate Z-Sampling's results under the uncertainty scheduler. Then, we analyze how this approximation error affects the performance of Z-Sampling.

Table 14: Quantitative comparison of Standard Sampling and Z-Sampling with the best grid searched guidance scale on Pick-a-Pic Dataset. Z-Sampling consistently outperforms across SDXL, SD2.1, and DreamShaper-xl-v2-turbo, indicating significantly better performance, even with grid searched results. Note that previous studies commonly use the defaulted guidance scale instead of fine-tuning it for diffusion models. This further demonstrated that the advantage of Z-Sampling is robust to various settings.

| Model | Method | HPS v2 ↑ | AES ↑ | PickScore ↑ | IR ↑ | Winning Rate ↑ |
|---|---|---|---|---|---|---|
| SD-2.1 | Standard | 26.86 | 5.70 | **20.39** | 23.87 | - |
| | Z-Sampling(ours) | **27.29** | **5.72** | 20.38 | **28.85** | 62% |
| SDXL | Standard | 31.00 | 6.10 | 21.72 | 79.17 | - |
| | Z-Sampling(ours) | **31.28** | **6.13** | **21.85** | **79.22** | 61% |
| DreamShaper -xl-v2-turbo | Standard | 30.16 | 5.99 | 21.53 | 68.99 | - |
| | Z-Sampling(ours) | **32.38** | **6.15** | **22.10** | **90.87** | 86% |

## E.1 UNCERTAINTY AND STOCHASTIC SAMPLERS

To assess the impact of different inversion algorithms on generation quality, we test various inversion methods. Specifically, we use SDXL-Turbo (4 steps) (Sauer et al., 2023) , an adversarial distillation diffusion model. Notably, SDXL-Turbo's default sampler is an ancestral Euler sampler, which introduces random noise at each denoising step, leading to highly inaccurate inversion.

Table 15: With stochastic samplers (e.g., Euler(a)), inversion inaccuracies reduce Z-Sampling's effectiveness. In contrast, deterministic samplers (e.g., Euler) yield better results with Z-Sampling.

| Method | HPS v2 ↑ | AES ↑ | PickScore ↑ | IR ↑ |
|---|---|---|---|---|
| Standard Sampling$_{\text{Euler(a)}}$ | **31.23** | **5.95** | 21.63 | **82.24** |
| Z-Sampling$_{\text{Euler(a)}}$ | 30.78 | 5.95 | **21.65** | 80.60 |
| Standard Sampling$_{\text{Euler}}$ | 27.05 | 5.60 | 20.36 | **41.44** |
| Z-Sampling$_{\text{Euler}}$ | **28.57** | **5.85** | **20.96** | 39.54 |

From Table 15, it can be seen that when using the Euler ancestral sampler, e.g., Euler(a), which introduces randomness in the denoising process, most metrics show a decline. This is because Euler(a) leads to inaccuracies in the inversion process, causing the approximation error term in equation 23 to increase significantly. As a result, Z-Sampling diverges from the data manifold, leading to reduced effectiveness.

However, when using deterministic Euler samplers, although the overall performance does not match that of the Euler(a) Sampler—acknowledging that other sampling methods on the turbo model may introduce blurring and related issues—Z-Sampling still demonstrates performance improvements over the corresponding baseline. For example, the PickScore increase from 20.3643 to **20.9639** This highlights the importance of the inversion algorithm and presents opportunities for improving Z-Sampling under stochastic samplers

Corresponding to equation 10, a deterministic sampler implies that the inversion process is imprecise, leading to an increase in $\tau_2(t)$. We note that end-to-end inversion amplifies the approximation error (Mokady et al., 2023), risking latents deviating from the data manifold. Z-Sampling, on the other hand, truncates the error at each step, reducing $\tau_2$, making semantic injection more efficient.

## E.2 THE INCREASE IN APPROXIMATION ERROR RESULTS IN NEGATIVE GAINS

To focus solely on the approximation error $\tau_2$ in Equation 10, we need to eliminate the influence of the semantic term $\tau_1$. So we set $\gamma_1 = \gamma_2 = 5.5$, which means $\delta_\gamma = 0$ and $\tau_1 = 0$. Then Equation 10

can be transformed as

$$\delta_{\text{Z-Sampling}} = \sum_{t=1}^{T}(x_t - \tilde{x}_t)^2 = \sum_{t=1}^{T}\alpha_t h_t^2 (\underbrace{\epsilon_\theta^t(\tilde{x}_t) - \epsilon_\theta^t(\tilde{x}_{t-1})}_{\tau_2(t):\text{approx error term}})^2. \tag{12}$$

Similarly, Equation 9 can be transformed as

$$\delta_{end2end} = (x_T - \tilde{x}_T)^2 = \alpha_T(\sum_{t=1}^{T} h_t (\underbrace{\epsilon_\theta^t(\tilde{x}_t) - \epsilon_\theta^t(\tilde{x}_{t-1})}_{\tau_2(t):\text{approx error term}}))^2. \tag{13}$$

Since the semantic term $\tau_1$ no longer contributes, only the effect of $\tau_2$ remains, as shown in Table 16 and Figure 26, both the end-to-end and step-by-step approaches result in negative gains. Notably, the approximation error introduced by the end-to-end method is two orders of magnitude higher than that of the step-by-step method, significantly degrading the image quality. This demonstrates that:

- An increase in the error term $\tau_2$ degrades the sampling effect.
- The step-by-step approach helps reduce the error term $\tau_2$, mitigating this negative gain.

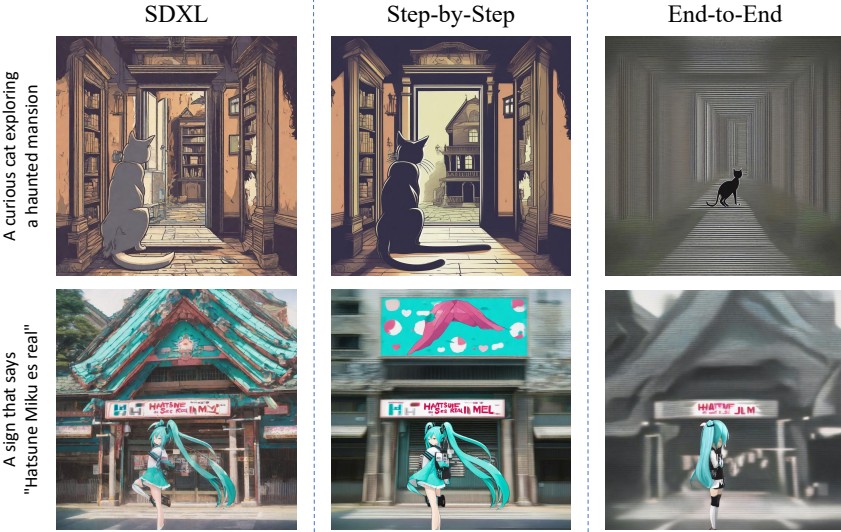

Figure 26: When the semantic term $\tau_1$ is removed (e.g., $\tau_1 = 0$), the presence of only the error term $\tau_2$ degrades the quality of generation results, and this negative gain effect is more pronounced in the end-to-end method.

Additionally, we test the performance of end-to-end and step-by-step methods in the presence of the semantic term $\tau_1$, as shown in Table 17. Since in this case, $\tau_1$ and $\tau_2$ are mixed together, so we only report the PickScore to reflect the quality of the generated results, as we are unable to report the exact Approx Error. It can be observed that with the presence of the semantic term, both methods yield positive gains, and the step-by-step method performs better.

Table 16: The results on Pick-a-Pick, excluding semantic term $\tau_1$. Model: SDXL.

| Method | $\delta_\gamma$ | PickScore ↑ | Approx Error $\tau_2$ |
|---|---|---|---|
| SDXL | - | **21.63** | 0 |
| End-to-End | 0 | 18.82 | 160.3313 |
| Step-by-Step | 0 | 21.52 | 0.9919 |

Table 17: The results on Pick-a-Pick, including semantic term $\tau_1$. Model: SDXL.

| Method | $\delta_\gamma$ | PickScore ↑ |
|---|---|---|
| SDXL | - | 21.63 |
| End-to-End | 5.5 | 21.65 |
| Step-by-Step | 5.5 | **21.85** |

### E.3 ARTIFICIALLY INTRODUCING GAUSSIAN ERROR

Specifically, to further illustrate that the approximation error $\tau_2$ leads to negative gains, we consider adding an additional random Gaussian term $\textbf{error}_{gs}$ to Equation 12, artificially simulating and controlling the inversion approximation error as

$$\delta_{\text{Z-Sampling}} = \sum_{t=1}^{T}(x_t - \tilde{x}_t)^2 = \sum_{t=1}^{T} \alpha_t h_t^2 (\underbrace{\epsilon_\theta^t(\tilde{x}_t) - \epsilon_\theta^t(\tilde{x}_{t-1})}_{\tau_2(t):\text{approx error term}} + s * \frac{norm(\epsilon_\theta^t(x_t))}{norm(\textbf{error}_{gs})}\textbf{error}_{gs})^2, \quad (14)$$

where $s$ is used to control the magnitude of the error. As seen in Table 18, the larger the value of s, the worse the performance of Z-Sampling, further illustrating that reducing the error term introduced by inversion is a direction that warrants attention.

Table 18: As the coefficient of the Gaussian error term increases, the quality of generation decreases.

| s | HPS v2 ↑ | AES ↑ | PickScore ↑ | IR ↑ |
|---|---|---|---|---|
| 0 | **29.95** | **6.1889** | **21.53** | **51.12** |
| 0.5 | 29.93 | 6.15 | 21.51 | 45.53 |
| 1.0 | 28.12 | 6.01 | 20.78 | 28.74 |

## F PROOFS

In this section, we derive the relationship between the end-to-end semantic injection approach and Z-Sampling, proving Z-Sampling's superiority. Then we formalize how Z-Sampling injects semantics via the guidance gap.

**Proof F.1 (Theorem 1)** *Given inference timesteps of $T$, from equation 4, we can obtain the inverted latent $\tilde{x}_T$ as*

$$\tilde{x}_T = \sqrt{\frac{\alpha_T}{\alpha_{T-1}}}\tilde{x}_{T-1} + \sqrt{\alpha_T}\left(\sqrt{\frac{1}{\alpha_T} - 1} - \sqrt{\frac{1}{\alpha_{T-1}} - 1}\right)\epsilon_\theta^T(\tilde{x}_{T-1}). \quad (15)$$

*For the sake of convenience, we set*

$$m_T = \sqrt{\frac{\alpha_T}{\alpha_{T-1}}}, \qquad n_T = \sqrt{\alpha_T}\left(\sqrt{\frac{1}{\alpha_T} - 1} - \sqrt{\frac{1}{\alpha_{T-1}} - 1}\right). \quad (16)$$

*So, equation 15 could also be written as*

$$\tilde{x}_T = m_T\tilde{x}_{T-1} + n_T\epsilon_\theta^T(\tilde{x}_{T-1}). \quad (17)$$

*Through iterative and combinatorial processes in equation 3, $\tilde{x}_T$ could be expressed as*

$$\begin{aligned}
\tilde{x}_T &= m_T\tilde{x}_{T-1} + n_T\epsilon_\theta^T(\tilde{x}_{T-1}) \\
&= m_Tm_{T-1}\tilde{x}_{T-2} + m_Tn_{T-1}\epsilon_\theta^{T-1}(\tilde{x}_{T-2}) + n_T\epsilon_\theta^T(\tilde{x}_{T-1}) \\
&= m_Tm_{T-1}m_{T-2}\tilde{x}_{T-3} + m_Tm_{T-1}n_{T-2}\epsilon_\theta^{T-2}(\tilde{x}_{T-3}) + m_Tn_{T-1}\epsilon_\theta^{T-1}(\tilde{x}_{T-2}) + n_T\epsilon_\theta^T(\tilde{x}_{T-1}) \\
&= \prod_{i=0}^{T}m_i\tilde{x}_0 + \sum_{t=1}^{T}n_t\prod_{k=t+1}^{T}m_k\epsilon_\theta^t(\tilde{x}_{t-1}). \quad (18)
\end{aligned}$$

*Similarly, based on equation 1 and equation 2, we can perform iterative derivations to obtain the equivalent form of $x_T$ as*

$$x_T = \prod_{i=0}^{T} m_i x_0 + \sum_{t=1}^{T} n_t \prod_{k=t+1}^{T} m_k \epsilon_\theta^t(x_t). \tag{19}$$

*We can determine the difference between $x_T$ and $\tilde{x}_T$, representing the gain from end-to-end semantic injection as*

$$
\begin{aligned}
\delta_{end2end} &= (x_T - \tilde{x}_T)^2 \\
&= \left( \prod_{i=0}^{T} m_i (x_0 - \tilde{x}_0) + \sum_{t=1}^{T} n_t \prod_{k=t+1}^{T} m_k \left( \epsilon_\theta^t(x_t) - \epsilon_\theta^t(\tilde{x}_{t-1}) \right) \right)^2 \\
&= \left( \sum_{t=1}^{T} \sqrt{\alpha_T} \left( \sqrt{\frac{1}{\alpha_t} - 1} - \sqrt{\frac{1}{\alpha_{t-1}} - 1} \right) \left( \epsilon_\theta^t(x_t) - \epsilon_\theta^t(\tilde{x}_{t-1}) \right) \right)^2 \\
&= \alpha_T \left( \sum_{t=1}^{T} \left( \sqrt{\frac{1}{\alpha_t} - 1} - \sqrt{\frac{1}{\alpha_{t-1}} - 1} \right) \left( \epsilon_\theta^t(x_t) - \epsilon_\theta^t(\tilde{x}_{t-1}) \right) \right)^2,
\end{aligned}
\tag{20}
$$

*where we set $h_t = \frac{n_t}{\sqrt{\alpha_t}}$, and further refine equation 20 to yield the semantic injection term $\tau_1$ and the approximation error term $\tau_2$ as*

$$
\begin{aligned}
\delta_{end2end} &= \alpha_T \left( \sum_{t=1}^{T} h_t \left( \epsilon_\theta^j(x_t) - \epsilon_\theta^t(\tilde{x}_t) \right) \right)^2 \\
&= \alpha_T \left( \sum_{t=1}^{T} h_t \left( \underbrace{\epsilon_\theta^t(x_t) - \epsilon_\theta^t(\tilde{x}_t)}_{\tau_1 : \text{semantic information gain term}} + \underbrace{\epsilon_\theta^t(\tilde{x}_t) - \epsilon_\theta^t(\tilde{x}_{t-1})}_{\tau_2 : \text{approx error term}} \right) \right)^2.
\end{aligned}
\tag{21}
$$

**Proof F.2 (Theorem 2)** *Unlike end-to-end approaches, in Z-Sampling, we focus solely on the local cycle of "$x_t \to x_{t-1} \to \tilde{x}_t$". Substituting equation 2 into equation 4 yields $\tilde{x}_t$ as*

$$
\begin{aligned}
\tilde{x}_t &= x_t - \sqrt{1 - \alpha_t} \epsilon_\theta^t(x_t) + \sqrt{\frac{(1 - \alpha_{t-1})\alpha_t}{\alpha_{t-1}}} \epsilon_\theta^t(x_t) \\
&\quad + \sqrt{\alpha_t} \left( \sqrt{\frac{1}{\alpha_t} - 1} - \sqrt{\frac{1}{\alpha_{t-1}} - 1} \right) \epsilon_\theta^t(\tilde{x}_{t-1}) \\
&= x_t + \sqrt{1 - \alpha_t} \left( \epsilon_\theta^t(\tilde{x}_{t-1}) - \epsilon_\theta^t(x_t) \right) + \sqrt{\frac{(1 - \alpha_{t-1})\alpha_t}{\alpha_{t-1}}} \left( \epsilon_\theta^t(x_t) - \epsilon_\theta^t(\tilde{x}_{t-1}) \right) \\
&= x_t + \left( \sqrt{1 - \alpha_t} - \sqrt{\frac{(1 - \alpha_{t-1})\alpha_t}{\alpha_{t-1}}} \right) \left( \epsilon_\theta^t(x_t) - \epsilon_\theta^t(\tilde{x}_{t-1}) \right) \\
&= x_t + \sqrt{\alpha_t} \left( \sqrt{\frac{1}{\alpha_t} - 1} - \sqrt{\frac{1}{\alpha_{t-1}} - 1} \right) \left( \epsilon_\theta^t(x_t) - \epsilon_\theta^t(\tilde{x}_{t-1}) \right).
\end{aligned}
\tag{22}
$$

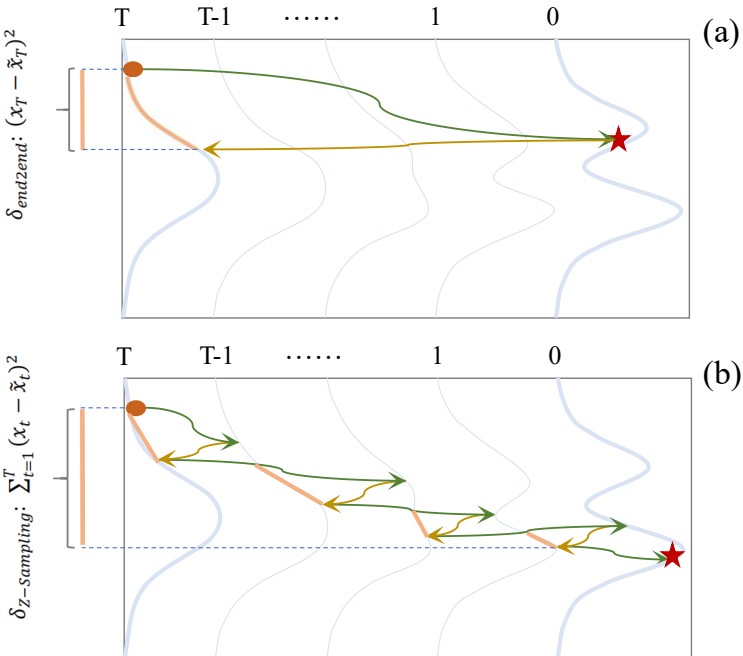

Figure 27: The End-to-End injection risks semantic cancellation across stages, leading to suboptimal results. In contrast, Z-Sampling captures and injects semantic information at each step in a timely manner along the sampling path, resulting in a stronger injection effect.

*The latent difference of Z-Sampling is accumulated as*

$$\delta_{\text{Z-Sampling}} = \sum_{t=1}^{T} (x_t - \tilde{x}_t)^2$$

$$= \sum_{t=1}^{T} \alpha_t h_t^2 \left( \epsilon_\theta^t(x_t) - \epsilon_\theta^t(\tilde{x}_{t-1}) \right)^2$$

$$= \sum_{t=1}^{T} \alpha_t h_t^2 \left( \underbrace{\epsilon_\theta^t(x_t) - \epsilon_\theta^t(\tilde{x}_t)}_{\tau_1 : \text{semantic information gain term}} + \underbrace{\epsilon_\theta^t(\tilde{x}_t) - \epsilon_\theta^t(\tilde{x}_{t-1})}_{\tau_2 : \text{approximation error term}} \right)^2. \quad (23)$$

In Figure 27, we visually represent the effect of equation 21 and equation 23. Z-Sampling clearly injects semantic information at each step in a timely manner, leading to a more pronounced effect and a deeper level of semantic injection.

We note in Equation 24 that $\epsilon_\theta^t(\tilde{x}_t)$ actually represents the denoising result of latent $x_t$ under low guidance $\gamma_2$, written this way for consistency with Equation 5. Therefore, the only difference between $\epsilon_\theta^t(\tilde{x}_t)$ and $\epsilon_\theta^t(x_t)$ is the guidance scale: $\epsilon_\theta^t(x_t)$ uses the guidance scale of $\gamma_1$, while $\epsilon_\theta^t(\tilde{x}_t)$ uses the guidance scale of $\gamma_2$. The latent input to the denoising network is the same for both $x_t$.

**Proof F.3 (Theorem 3)** *Excluding the approximation error introduced by inversion algorithm, we can rewrite equation 23 as*

$$\delta_{\text{Z-Sampling}} = \sum_{t=1}^{T} \alpha_t h_t^2 \left( \epsilon_\theta^t(x_t) - \epsilon_\theta^t(\tilde{x}_t) \right)^2. \quad (24)$$

*Although the step-by-step approach results in $x_t$ and $\tilde{x}_t$ being the same at each timestep $t$, from equation 5, we note that $\epsilon_\theta^t(x_t)$ and $\epsilon_\theta^t(\tilde{x}_t)$ are obtained under guidance scales $\gamma_1$ and $\gamma_2$ respec-*

*tively. Thus, the effect of Z-Sampling is further equivalent as*

$$
\begin{aligned}
\delta_{\text{Z-Sampling}} &= \sum_{t=1}^{T} \alpha_t h_t^2 \left( (\gamma_1 - \gamma_2) \, u_\theta(x_t, c, t) - (\gamma_1 - \gamma_2) \, u_\theta(x_t, \varnothing, t) \right)^2 \\
&= \sum_{t=1}^{T} \alpha_t h_t^2 \left( (\gamma_1 - \gamma_2) \left( u_\theta(x_t, c, t) - u_\theta(x_t, \varnothing, t) \right) \right)^2 \\
&= \sum_{t=1}^{T} \alpha_t h_t^2 \left( \delta_\gamma \left( u_\theta(x_t, c, t) - u_\theta(x_t, \varnothing, t) \right) \right)^2 .
\end{aligned}
\tag{25}
$$

*Here, $\delta_\gamma$ represents the guidance gap between denoising and inversion, i.e., $\gamma_1 - \gamma_2$.*

From equation 25, we note that the effectiveness of Z-Sampling primarily depends on:

1. The guidance gap $\delta_\gamma$, which we can control to regulate the magnitude and intensity of the optimization.

2. The difference between the conditional branch $u_\theta(x_t, c, t)$ and unconditional branch $u_\theta(x_t, \varnothing, t)$, which is determined by the prompt c and the model parameters $\theta$.

As mentioned in the end of Proof F.2, in the absence of inversion approximate errors, the only difference between $\epsilon_\theta^t(x_t)$ and $\epsilon_\theta^t(\tilde{x}_t)$ in Equation 24 is they use the different guidance scale. Therefore, even when $\gamma_2 = 0$, our focus remains on the invariant, which is the difference between the network outputs of the conditional and unconditional branches $u_\theta(x_t, c, t) - u_\theta(x_t, \varnothing, t)$.

## G  THE END-TO-END SEMANTIC INJECTION ALGORITHM

In this section, we show how to inject semantic information end-to-end as described in Section 3.3.

---

**Algorithm 2** End-to-End Semantic Injection

---

1: **Input:** Denoising Process: $\Phi$, Inversion Process: $\Psi$, text prompt: $c$, denoising guidance: $\gamma_1$, inversion guidance: $\gamma_2$, inference steps: $T$
2: **Output:** Clean image $x_0$
3: Sample Gaussian noise $x_T$
4: $x_0 = \phi(x_T | c, \gamma_1)$     #see equation 6
5: $\tilde{x}_T = \psi(x_0 | c, \gamma_2)$     #see equation 7
6: $x_0 = \phi(\tilde{x}_T | c, \gamma_1)$     #see equation 8
7: **return** $x_0$

---

