# OpenReview forum: "Zigzag Diffusion Sampling: Diffusion Models Can Self-Improve via Self-Reflection"
_ICLR.cc/2025/Conference — ICLR 2025 Poster_

### Official Review · Reviewer_9XhW · 2024-10-16

**Soundness:** 3
**Presentation:** 3
**Contribution:** 3
**Rating:** 6
**Confidence:** 3

**Summary:**

The paper presents a novel sampling method, Zigzag Diffusion Sampling (Z-Sampling), designed to enhance the generation quality of text-to-image diffusion models. The key contribution lies in leveraging the guidance gap between the denoising and inversion processes to accumulate semantic information throughout the generation process. This leads to improved image quality and prompt-image alignment, especially for challenging prompts. Z-Sampling is a plug-and-play method that can be easily integrated into various diffusion models. Extensive experiments demonstrate its superior performance across different datasets, models, and evaluation metrics.

**Strengths:**

- Clear and well-organized paper: The paper is well-structured, making it easy to follow the authors' arguments.
- Theoretical and empirical analysis: The authors provide a strong theoretical foundation for the guidance gap between the denoising and inversion processesand  their approach and support it with extensive empirical evidence.
-  Z-Sampling is a unique contribution to the field of diffusion models, offering a new way to improve generation quality.
- Plug-and-play nature: The method's simplicity and ease of integration make it attractive for researchers and practitioners.
- Strong experimental results: The authors demonstrate the effectiveness of Z-Sampling through comprehensive experiments.

**Weaknesses:**

- Comparison to other training-free enhancement techniques: While the paper mentions other training-free enhancement techniques, a more in-depth comparison with methods like SAG[1], SEG[2], and CFG++[3] would strengthen the paper. This would provide a clearer understanding of Z-Sampling's unique advantages.
- Comparision to "Text-to-Image Rectified Flow as Plug-and-Play Priors"[4]: The proposed method has similarity with this paper in terms of utilizing the forwad and reverse process to enhance the quality.


[1] Hong, Susung, et al. "Improving sample quality of diffusion models using self-attention guidance." Proceedings of the IEEE/CVF International Conference on Computer Vision. 2023.

[2] Hong, Susung. "Smoothed Energy Guidance: Guiding Diffusion Models with Reduced Energy Curvature of Attention." arXiv preprint arXiv:2408.00760 (2024).

[3] Chung, Hyungjin, et al. "CFG++: Manifold-constrained Classifier Free Guidance for Diffusion Models." arXiv preprint arXiv:2406.08070 (2024).

[4] Yang, Xiaofeng, et al. "Text-to-Image Rectified Flow as Plug-and-Play Priors." arXiv preprint arXiv:2406.03293 (2024).

**Questions:**

- Would you compare Z-sampling with other training-free enhancement techniques?
- Would you compare Z-sampling with "Text-to-Image Rectified Flow as Plug-and-Play Priors"?

---

> ### Author Response · Authors · 2024-11-23
> **Response to Reviewer 9XhW (Part 1/1)**
>
> We appreciate the reviewer 9XhW for the thorough feedback, and are delighted that our work has received your recognition. We concur with reviewer's suggestion to compare with more methods and reference/discuss them in the paper.  Below are the experimental results.
>
> ---
>
> **Q1:** Would the author compares Z-sampling with other training-free enhancement techniques?
>
> **A1:** Thanks for the suggestion. Yes, we would like to. We present the results compared across multiple baselines below, and we report more details in **Table 9** and **Table 10 of Appendix D.1**. It further supports that Z-Sampling can still outperform more recent baselines.
>
> - SAG[1]/SEG[2]
>
>   We note that SAG[1] employs blur guidance and intermediate self-attention maps to achieve higher quality samples. Building upon this, SEG[2] further optimizes this method from the energy landscape perspective. We cite both of them in revision. Since SEG is a more recent powerful method, we report the comparison results with SEG in Pick-a-Pic Dataset given the rebuttal time.
>
>   In SEG, its characteristic is that it can generate well under unconditional conditions without specifying a default cfg guidance scale. Therefore, we adopt the default setting in our paper (i.e., cfg guidance scale \gamma_{1} = 5.5). We found that Z-Sampling outperforms SDXL and SEG.
>
>   | (Pick-a-Pic) | HPS v2↑ | AES↑ | PickScore↑ | IR↑ | (DrawBench) | HPS v2↑ | AES↑ | PickScore↑ | IR↑ |
>   |---------|---------|---------|---------|---------|---------|---------|---------|---------|---------|
>   | SDXL   | 0.2989  |  6.0870 |  21.6353 |  0.5865 | SDXL | 0.2881  |  5.5595 |  22.3086 |  0.6075 |
>   | SEG   | 0.3053   | 6.1231  | 21.4186 | 0.6157 | SEG | 0.2960  |  5.6596 |  22.1453 |  0.6042 |
>   | Ours   | **0.3128**  | **6.1302**  | **21.8477** | **0.7922** | Ours | **0.3050** | **5.6739** | **22.4581** | **0.7997** |
>
> ---
>
> - CFG++[3]
>
>   As for CFG++[3], since it restricts the cfg scale to the range [0,1], while the classic Z-Sampling is larger, such as the range [3,10], a fair comparison is not possible. Given this, we use $\omega = 0.5$ in CFG++, corresponding to a cfg scale of 5.5 in Z-Sampling. (Refer to Table 1 in [3]).
>
>   It is worth noting that in the official implementation of CFG++, the VAE encoder uses "**madebyollin/sdxl-vae-fp16-fix**" checkpoint. For fair comparison, we follow this setting, so the results reported for SDXL and Z-Sampling are slightly different from the previous results.
>
>   | (Pick-a-Pic) | HPS v2↑ | AES↑ | PickScore↑ | IR↑ | (DrawBench) | HPS v2↑ | AES↑ | PickScore↑ | IR↑ |
>   |---------|---------|---------|---------|---------|---------|---------|---------|---------|---------|
>   | SDXL   | 0.3004  |  6.1121 |  21.8053 |  0.6007 | SDXL | 0.2885  |  5.6245 |  22.4213 |  0.6761 |
>   | CFG++  | 0.3028 | 6.0989 | 21.8337 | 0.6730 | CFG++ | 0.2865 |  5.6174|  22.3797 |  0.6266 |
>   | Ours   | **0.3124**  | **6.1170**  | **21.8444** | **0.7855** | Ours | **0.3035** | **5.6594** | **22.4392** | **0.7911** |
>
> ---
>
> **Q2:** Would the author compares Z-sampling with the training mathods "Text-to-Image Rectified Flow as Plug-and-Play Priors"[4]?
>
> **A2:** Thanks for the reviewer's suggestion. Due to RFDS's need for a pretrained flow model and latent space training, comparing it with Z-Sampling, is challenging. Specifically, we note that
>
> - RFDS
>
>   A method utilizing fixed prior weights $\phi$ to assist in the training of learnable parameters $\theta$.
>
> - IRFDS
>
>    A method utilizing fixed prior weights $\phi$ to assist in the training of learnable noise $\epsilon$.
>
> - RFDS-REV
>
>   Combining RFSD and iRFSD, i.e., optimizing noise $\epsilon$ before updating parameters  $\theta$, to enhance generation quality.
>
> RFSD and RFDS-REV improve the training process, and iRFSD enhances image editing performance (Table 2 of [4]). We believe that iRFSD, as a method for optimizing noise based on training, can reduce the approximation error of inversion in Z-Sampling. Thanks to the reviewer's suggestion, **we also discuss this point in Appendix B**.
>
> ---
>
> We hope that these additional experiments and explanations resolve the reviewer’s concerns and questions. We are more than happy to discuss if the reviewer  9XhW has further questions or comments.
>
> ---
>
> [1] Hong, Susung, et al. "Improving sample quality of diffusion models using self-attention guidance." Proceedings of the IEEE/CVF International Conference on Computer Vision. 2023.
>
> [2] Hong, Susung. "Smoothed Energy Guidance: Guiding Diffusion Models with Reduced Energy Curvature of Attention." The Thirty-eighth Annual Conference on Neural Information Processing Systems.
>
> [3] Chung, Hyungjin, et al. "CFG++: Manifold-constrained Classifier Free Guidance for Diffusion Models." arXiv preprint arXiv:2406.08070 (2024).
>
> [4] Yang, Xiaofeng, et al. "Text-to-Image Rectified Flow as Plug-and-Play Priors." arXiv preprint arXiv:2406.03293 (2024).

---

> > ### Comment · Reviewer_9XhW · 2024-11-26
> >
> > Thank you for your thoughtful response and the updated results. I will keep my score unchanged.

---

### Official Review · Reviewer_BMZe · 2024-10-31

**Soundness:** 3
**Presentation:** 3
**Contribution:** 2
**Rating:** 6
**Confidence:** 5

**Summary:**

The paper introduces a novel approach called Zigzag Diffusion Sampling (Z-Sampling) to enhance the performance of existing diffusion models, particularly for complex and fine-grained prompts in text-to-image generation tasks. This method employs a zigzag-like alternating denoising and inversion process, which progressively accumulates semantic information. Z-Sampling enables flexible control over the injection of semantic information, is applicable across various diffusion models, and requires minimal code modifications. Extensive validation on multiple benchmark datasets, diffusion models, and evaluation metrics demonstrates its effectiveness, especially in handling complex prompts such as style, position, counting, and multiple objects. As a training-free approach, Z-Sampling significantly improves baseline performance with lower computational cost, showing substantial practical value.

**Strengths:**

- Through zigzag sampling, semantic information in the latent space is progressively accumulated, effectively enhancing the alignment between generated content and prompts, especially for complex prompts.

- Z-Sampling is a plug-and-play method that can be seamlessly applied to different types of diffusion models with minimal code modifications.

- As a training-free sampling method, Z-Sampling reduces computational costs while maintaining high generation quality, with experiments demonstrating its performance superiority over standard sampling within the same time frame.

**Weaknesses:**

- The authors’ insight—that the semantic information contained in the latent affects generation results—is quite evident.
- The method’s process is very simple and straightforward, which reminds me of time travel. The difference is that, in time travel, noise is added when tracing back, while zigzag uses inversion. The improvement in time travel comes from the inherent bias in the sampling process. By performing additional noise addition and denoising steps, the distribution can be pulled back onto the intermediate data manifold, thereby improving quality. Thus, to imbue $x_t$ with more semantic information of $c$, using inversion instead of time travel’s random noise to pull the data distribution back onto the intermediate data manifold $p_t(x_t|c)$ for enhancing conditional sampling quality seems like an obvious approach.

**Questions:**

- See Weakness
- In each zigzag optimization, only one step of denoising and one step of inversion is performed. Have you considered performing multiple steps of denoising and multiple steps of inversion?
- I am also curious about Figure 9. Theoretically, using the zigzag algorithm indeed quick and effective alignment, but as the number of optimization steps approaches infinity, the accumulated prediction errors should lead to a gradual downward trend. However, Figure 9 does not reflect this; it even shows an accelerating upward trend. How can this phenomenon be explained?

---

> ### Author Response · Authors · 2024-11-23
> **Response to Reviewer BMZe (Part 1/2)**
>
> We appreciate the reviewer BMZe for the thoughtful feedback. We are glad to hear that the reviewer appreciates the usability and general applicability of Z-Sampling. We hope these discussions can further clarify and elaborate on the contributions of Z-Sampling.
>
> We duly address the reviewer BMZe’s all concerns below.
>
> ---
>
> **Q1:** The authors’ insight—that the semantic information contained in the latent affects generation results—is quite evident.
>
> **A1:** We first thank the reviewer for acknowledging the point that "the semantic information contained in the latent affects generation results," which means our empirical observations (see Sec 3.1) have been reasonably recognized.
>
>  It is important to emphasize that, prior to our work, no research had identified that each step of the zigzag operation can accumulate the information gain between strong guidance and weak guidance.   And we are the first to explicitly articulate this concept, offer a credible theoretical explanation for it, and substantiate it with extensive empirical evidence.
>
> Here, we want to outline the insights from our work andhope to clarify our contributions again:
>
> - **What signifies a better latent variable in the Diffusion Model?** (Sec 3.1)
>
>   Observing the latent's generation results under weak conditions to judge its performance under specific strong conditions.
>
> - **How to improve the properties of an initial latent variable to enhance generation quality?**  (Sec 3.2)
>
>   Demonstrating that the gap between cfg scales is a key component, controlling this gap can determine the direction and intensity of optimization.
>
> - **Why end-to-end methods is inferior to step-by-step approaches?** (Sec 3.3)
>
>   Conducting theoretical and empirical analysis to reveal the impact of semantic term $\tau _{1}$ and error terms $\tau _{2}$, further improving the algorithm.
>
> We understand the reviewer's concern. While it seems that insight #1 is a reasonable observation, we believe that deriving subsequent conclusions from this observation is inspiring and interesting work, including investigating the classifier guidance gap, capturing semantics based on this observation, or deriving inversion errors.
>
> ---
>
> **Q2:** Z-Sampling and the time-travel approach seem somewhat similar.
>
> **A2:**  We respectfully note that we have introduced this method (termed as Re-sampling) in related work section and compared it with our approach. This method was first introduced in Repaint [1], where it was termed Resampling. Subsequently, Freedom [2] adopted this approach, referring to it as Time-Travel. In essence, both Resampling and Time-Travel are the same algorithm.
>
> Here, we emphasize the differences between Z-Sampling and Resampling.
>
> - **Empirically**, we provide a extensive comparison between Z-Sampling and Resampling in Table 1 and Table 2. Z-Sampling consistently outperforms Resampling across nearly  all metrics (such as HPS v2, PickScore, etc.) on all benchmarks.
>
> - **Theoretically**, Z-Sampling is supported by a clear theoretical foundation, which involves continuously accumulating semantic information from the gap between strong guidance and weak guidance. This allows us to flexibly control the direction and intensity of optimization, as well as provide guidance for further algorithm optimization. In contrast, Resampling has been mostly treated as a minor performance enhancement module in previous works, such as...
>
>   - In **Repaint** [1],  they provided a hypothetical description of less than a page (Sec 4.2 in [1]) for the specific task of repaint. There were no detailed experiments and theoretical analysis to explain why Resampling works; and it was merely treated as a **performance-enhancing trick**.
>
>   - In **Time Travel** [2], they also allocated less than one page to introduce Time Travel (Sec 4.2 in [2]), merely stating that "... it has **empirically** shown to inhibit the generation of disharmonious results...".
>
>   Why it works theoretically, how to develop better algorithms, and what controls the effect (e.g., guidance gap) — these aspects have received scant attention in previous work.
>
> Specifically, we believe in the importance of uncovering the underlying mechanisms—we earnestly hope that Reviewer BMZe also agrees with the significance of this endeavor.  And we organize the relationships with other works in **Appendix B**, hoping this could further clarify the contributions of our work.
>
> ---
> [1] Lugmayr, Andreas, et al. "Repaint: Inpainting using denoising diffusion probabilistic models." 2022.
>
> [2] Yu, Jiwen, et al. "Freedom: Training-free energy-guided conditional diffusion model."  2023.

---

> > ### Author Response · Authors · 2024-11-23
> > **Response to Reviewer BMZe (Part 2/2)**
> >
> > **Q3:** How do multiple denoising and inversion steps fare in Z-Sampling?
> >
> > **A3:**  Thanks for the reviewer's suggestions. We implement multiple steps in each zigzag operation, i.e., denoising for k steps  and then inversion for k steps with  k ranging from 1 to 4.
> >
> > We note that the best performance is achieved when k=1, the standard setting of Z-Sampling. **As k increases, the performance of Z-Sampling deteriorates**. This can be explained by Theorem 1 and Theorem 2, which demonstrated that k=1 can better accumulate the information gain.
> >
> > | **k** | HPS v2↑ | AES↑ | PickScore↑ | IR↑ |
> > |---------|---------|---------|---------|---------|
> > | 0 (e.g., SDXL)  | 0.2989   | 6.0870  | 21.6353  | 0.5865 |
> > | 1  | **0.3128**   | **6.1302**  | **21.8477**  | 0.7922 |
> > | 2  | 0.3111  | 6.0764  | 21.7163 | **0.8453** |
> > | 3   | 0.3075  | 6.0885  | 21.4848 | 0.7854 |
> > | 4   | 0.3059  | 6.0940  | 21.3357 | 0.7860 |
> >
> > Specifically, in the extreme case where $k=T-1$, if we only perform the zigzag operation on the initial latent, this equates to end-to-end zigzag (Equation 9). And we provide a more detailed discussion in **Table 12 of Appendix D.1**.
> >
> > ---
> >
> > **Q4:** There is some confusion regarding the interpretation of Figure 9.
> >
> > **A4:** We apologize for the inadequate description of Figure 9. Here is a more detailed explanation:
> >
> > We conducted experiments on Pick-a-Pick using SDXL (50 steps), where $\lambda$ indicates the first $\lambda$ steps using the zigzag operation. For example, when $\lambda$ is 0, it reverts to standard SDXL. When $\lambda$ is 25, it means the first 25 steps of the denoising process use the zigzag operation.
> >
> > As shown in Figure 9, when $\lambda$ increases from 0 to 25, the winning rate rises from 50% to 75%. However, when increasing from 25 to 50 steps, it only rises from 75% to 80%. This indicates that the **zigzag operation is more effective during the early stages of denoising**.
> >
> > ---
> >
> > Finally, we sincerely thank the reviewer’s feedback. It definitely inspires us to further improve our work and clarification for more readers. We respectfully hope that the reviewer could reevaluate our work given the reponses addresing your main concerns. And we look forward to further discussion with the reviewer BMZe.

---

> > > ### Comment · Reviewer_BMZe · 2024-11-25
> > >
> > > I thank the authors for their thorough reply, the authors have answered my questions satisfyingly. I thus raise my score on the paper.

---

### Official Review · Reviewer_A4t2 · 2024-11-02

**Soundness:** 3
**Presentation:** 3
**Contribution:** 2
**Rating:** 6
**Confidence:** 4

**Summary:**

This paper identifies that the guidance gap between denoising and inversion captures prompt-specific semantic information and proposes Z-Sampling to gradually accumulate this information throughout the generation process, through iteratively denoising and reversing. Extensive experiments demonstrate that  Z-Sampling significantly improves image quality and prompt alignment, outperforming other baseline models.

**Strengths:**

1. The paper demonstrates strong problem identification and analysis, accurately pinpointing the reasons behind guidance failures with certain seeds in text-to-image diffusion models.

2. The presentation of the paper is clear and well-structured, making it easy to follow and understand the theoretical insights and proposed methods.

3. The experimental results are thorough, covering multiple baselines and benchmarks to robustly validate the effectiveness of the approach.

**Weaknesses:**

1. My major concern is that the qualitative results appear unsatisfactory. In almost all qualitative examples, Z-Sampling exhibits a pronounced **oversaturation** effect, with excessively vivid subject colors and significantly blurred backgrounds. While this approach may excel on certain benchmarks, these results are unlikely to be perceived as high-quality by users in practical evaluations.

2. Following from weakness 1, this oversaturated, unrealistic effect often occurs at very high CFG scales. As shown in Figure 3, Z-Sampling may essentially function as a high-CFG sampling method to enforce guidance. The paper does not provide a detailed comparison between Z-Sampling and high-CFG sampling, either theoretically or experimentally, and the results suggest that both approaches share similar drawbacks.

3. The paper lacks deeper theoretical analysis. The mathematical proofs provided rely on basic formula applications, but Z-Sampling alters the sampling path, creating a shift between the denoising trajectories of inference and training. In contrast, standard CFG sampling has rigorous theoretical backing. Revisiting the score function to analyze how training and inference could align under Z-Sampling conditions would strengthen the paper’s theoretical foundation.

4. Limited technical innovation. The proposed iterative inverse-and-denoise approach has already been used in various image editing applications, and also, in image to video methods like Tune-A-Video[1] to enhance structural coherence. The paper would benefit from further elaboration to clarify its unique technical contributions.

5. Lack of comprehensive analysis. CFG sampling incorporates not only semantic information but also an unconditional branch that helps with structural aspects. By attributing improvements solely to semantic response, the paper misses an analysis of structural and visual enhancements. A more detailed examination here could strengthen the paper’s argument and overall impact.

6. There are some typographical errors. For instance, in Algorithm 1, the steps for denoising and inversion are mistakenly reversed.

[1] Wu, Jay Zhangjie, et al. "Tune-a-video: One-shot tuning of image diffusion models for text-to-video generation." Proceedings of the IEEE/CVF International Conference on Computer Vision. 2023.

**Questions:**

1. In Tables 1 and 2 of the main results, what guidance scales were used for each model, and what was the specific setting for $\gamma_1$?
2. In Figure 10, how is the x-axis varied? My understanding is that the time taken for a model to generate a single image should be constant.

3. How might Z-Sampling be integrated into the training phase to achieve improved results? Currently, the oversaturation effect seems too pronounced.

4. $\gamma_2$ is set to 0 in most cases, suggesting that the reverse process disregards unconditional information entirely. Could a more detailed theoretical explanation be provided for this finding?

---

> ### Author Response · Authors · 2024-11-23
> **Response to Reviewer A4t2 (Part 1/4)**
>
> We appreciate the reviewer A4t2 for the thoughtful feedback, which has deepened our reflection on our work. Below we attempt to address each of the reviewer A4t2 's concerns.
>
> ---
>
> **Q 1:** The visual results reported in the paper appear to be oversaturated.
>
> **A 1:** We respectfully argue that the oversaturation is mild and is not responsible for the improvement in quantitative metrics for Z-Sampling. We note that **Z-Sampling could find a superior trade-off between image quality and prompt adherence**.
>
> First, we conduct experiments at an extremely high CFG scale (equivalent to three times the recommended guidance scale). In this scenario, standard sampling performs unquestionably poorly, exhibiting severe artifacts/oversaturation. However, using Z-Sampling, we are able to mitigate these issues to some extent (Please see **Figure 24** in the revision). For example, if saturation is too low, Z-Sampling increases it (as pointed out by the reviewer A4t2); if saturation is too high (typically occurring at excessively high CFG scale), Z-Sampling decreases it.
>
> Second, in Table 2, we report the performance of Z-Sampling on GenEval, a complex benchmark that includes metrics such as counting, position, and attribute binding, which are unrelated to saturation. We note that Z-Sampling still achieves better performance.
>
> ---
>
> **Q 2:** Z-Sampling seems to have similar effects to high CFG, sharing the same drawbacks.
>
> **A 2:**  We respectfully note that the effects of Z-Sampling are actually different from high CFG scales. **Figure 24** in the revision reveals this point. To further validate our claim, we also present the quantitative results under different CFG values in the table below (also shown in Table 13 in the revision). Due to deadline constraints, we use DreamShaper-xl-turbo-v2 as the base model.
>
> The standard sampling performs best at $\gamma_{1}=3.5$, which is also the official recommended guidance sclae. While $\gamma_{1}>3.5$, the performance of standard sampling begins to decline. However, Z-Sampling consistently yields signficant performance gains, indicating that our method can still work effectively under high guidance scales. Specifically, in **Figure 23**, we note that **even at extremely high CFG scale** (e.g., $\tau_{1}=11.5$), **the winning rate remains above 80%**.
>
> | Method | $\gamma_{1}$ | HPS v2↑ | AES↑ | PickScore↑ | IR↑ | winning rate↑ |
> |---|---|---|---|---|---|---|
> | Standard Sampling | 1.5 | 0.2851 | 5.8327 | 21.3729 | 0.4325 | - |
> | Z-Sampling | 1.5 | **0.2951** | **6.0143** | **21.6541** | **0.5589** |**73%**|
> | Standard Sampling | 3.5 | 0.3004 | 5.9355 | 21.5899 | 0.6618 | -
> | Z-Sampling | 3.5 | **0.3238** | **6.1542** | **22.1025** | **0.9087** | **88%**|
> | Standard Sampling | 5.5 | 0.2996 | 5.9668 | 21.3718 | 0.6446 | - |
> | Z-Sampling | 5.5 | **0.3142** | **6.0513** | **21.8309** | **0.7600** | **85%** |
> | Standard Sampling | 7.5 | 0.2910 | 5.8816 | 21.0236 | 0.6026 | - |
> | Z-Sampling | 7.5 | **0.3090** | **5.9537** | **21.5977** | **0.7418** | **86%** |
> | Standard Sampling | 9.5 | 0.2798 | 5.7649 | 20.5981 | 0.4170 | - |
> | Z-Sampling | 9.5 | **0.2995** | **5.8788** | **21.2806** | **0.6340** | **92%** |
> | Standard Sampling | 11.5 | 0.2693 | 5.6030 | 20.3055 | 0.3145 | - |
> | Z-Sampling | 11.5 | **0.2897** | **5.7694** | **20.9710** | **0.5569** | **91%** |
>
> ---

---

> ### Author Response · Authors · 2024-11-23
> **Response to Reviewer A4t2 (Part 2/4)**
>
> ---
>
> **Q3:** The theoretical analysis of Z-Sampling is insufficient, particularly lacking in analysis related to training.
>
> **A3:** We believe that the true value of a theoretical analysis depends on its novelty and the insight it provides for subsequent research.
>
> - **In terms of novelty**，we note that before us, there is no research realized that the gap between strong guidance and weak guidance could be used to study how this accumulated gain affects the generation results. We are the first to explicitly articulate this concept, offer a theoretical explanation, and substantiate it with extensive experiments.
>
> - **In terms of insights for subsequent research**, we thoroughly explored the impact of Z-Sampling's components on generation effects. Based on our theory, future work can be optimized in multiple directions, such as designing better accumulation methods for the semantic term $\tau_{1}$, exploring approaches to reduce the inversion error term, and applying this semantic accumulation gain to zero-shot discriminative tasks (such as image classification).
>
> Additionally, since our research focuses on the diffusion sampling algorithm, training is beyond the scope of this paper. As the reviewers noted, Z-Sampling holds significant potential in the training phase, which we will explore in future work.
>
> Finally, we clarify the theoretical contributions of our work:
>
> - **Theorems 1 and 2** show that the effect of Z-Sampling can be decomposed into the semantic term $\tau_{1}$ and the error term $\tau_{2}$.
>
> - **Theorem 3** unveils the pivotal aspect of this iterative inverse-and-denoise approach, emphasizing that the CFG guidance gap is indeed significant. Specifically, positive optimization can only be achieved when $\gamma_{1} - \gamma_{2}>0$, a perspective that has not been sufficiently explored.
>
> We hope the reviewer could understand the efforts we have made, which aim to bring more new thinking to the community through Z-Sampling and its core guidance gap mechanism. And we will continue this work in future researche, especially theoretical the analysis on how latent spaces possess prior semantic information.
>
> ---
>
> **Q4:** Limited technical innovation of Z-Sampling.
>
> **A4:** We thank the reviewer A4t2 for mentioning this work [2], and we will reference and discuss it in the revision.  We  note that Z-Sampling is the first work that has conducted a detailed analysis of the zigzag mechanism, including cfg guidance, error terms, semantic accumulation, and extensive ablation experiments to validate the correctness of the theory. In contrast, as a broad inference paradigm, most research simply treats it as a plug-and-play sub-module to improve performance**, for example:
>
> - In **Tune-a-video** [2], only a quarter of a page is devoted to discussing this iterative paradigm (Sec 3.3 in [2]). They just say "well, this seems to help stabilize the structural integrity, which is enough".  In fact, Sec 3.3 in  [2] just employs an end-to-end optimization approach (which has been proven suboptimal, see Appendix E.2 in the revision), and doesn't consider that further reducing the guidance scale during inversion might yield better performance.
>
> - In **Time Travel** [3], they also allocate less than a page to introduce time travel algorithm (Sec 4.2 in [3]), merely stating that "... it has empirically shown to inhibit the generation of disharmonious results...".
>
> We highly value this feedback, which has helped us clarify our contributions and refine the positioning of our work. We hope that Z-Sampling, can provide a  extensive study of this iterative paradigm and inspire the community with further optimization directions (such as better semantic accumulation methods or smaller inversion errors). And we also hope the reviewer can understand this point and look forward to further discussions on this topic.
>
> ---
>
> [1] Ho, Jonathan, and Tim Salimans. "Classifier-Free Diffusion Guidance." NeurIPS 2021.
>
> [2] Wu, Jay Zhangjie, et al. "Tune-a-video: One-shot tuning of image diffusion models for text-to-video generation." ICCV 2023.
>
> [3] Yu, Jiwen, et al. "Freedom: Training-free energy-guided conditional diffusion model."  ICCV 2023.

---

> ### Author Response · Authors · 2024-11-23
> **Response to Reviewer A4t2 (Part 3/4)**
>
> **Q5:** Our work lacks analysis of the role of the unconditional branch.
>
> **A5:**
> We respectfully argue that our analysis also sheds light on $u_{\theta}(x_{t},c,t)-u_{\theta}(x_{t},\varnothing,t)$, rather than just $u_{\theta}(x_{t},c,t)$ or $u_{\theta}(x_{t},\varnothing,t)$, as indicated in Theorem 3. Following the suggestion, we further discuss the the unconditional branch deeply in Appendix F of the revision. It shows that the role of the unconditional branch is to provide a base signal and decouple useful semantic information from the conditional branch (in the form of a differential). Notably, this differential itself is independent of $\gamma_{1}$ or $\gamma_{2}$ (as evident from the input to $u_{\theta}$), and the main role of $\gamma_{1}-\gamma_{2}$ is to amplify or attenuate the magnitude of this differential.
>
> In the case of neglecting the approximation error $\tau_{2}$, the effect of Z-Sampling can be considered as
>
> $\delta_{Z-sampling} = \sum_{t=1}^{T} \alpha_{t}h_{t}^{2}\left((\gamma_{1}-\gamma_{2})\left(u_{\theta}(x_{t},c,t)-u_{\theta}(x_{t},\varnothing,t)\right)\right)^{2}.$
>
> In this formula, even if $\gamma_{2}$ is set to zero, the difference $\delta_{\gamma} = \gamma_{1} - \gamma_{2}$ is not zero, which means that the uncondtional branch $u_{\theta}(x_{t},\varnothing,t)$  still plays a role in this game,  and $u_{\theta}(x_{t},\varnothing,t)$ is provided by the denoising process rather than the inversion process in this scenario.  By controlling the magnitude and sign of $\delta_{\gamma}$, we determine the effect of the difference between $u_{\theta}(x_{t},c,t)$ and $u_{\theta}(x_{t},\varnothing,t)$. It is noted that in the ideal case, i.e., neglecting the inversion approximation error, even if $\gamma_{1}$ and $\gamma_{2}$ change, this difference $u_{\theta}(x_{t},c,t)-u_{\theta}(x_{t},\varnothing,t)$ remains unchanged.
>
> Additionally, it is important to note that our experiments is not limited to the setting where $\gamma_{2}=0$. In Figure 7/8, **we conducted experiments with different inversion guidance $\gamma_{2}$**. For example, when $\gamma_{2}>0$, the unconditional branch participates in the process, and $\delta_{\gamma}$ becomes smaller, reducing the gains of Z-Sampling. We note $\gamma_{2}=0$  is just a hyperparameter for main experiments after considering multiple metrics (such as HPS v2, AES, etc.), and it is not a special value.
>
> ---
>
> **Q6:** There is a typo error in the algorithm 1.
>
> **A6:** Thanks for the reviewer's feedback. However, after a thorough check, we think there are no typographical errors in Algorithm 1. We are afarid that the reviewer may misundertand the method a little bit.
>
> In Algorithm 1, at each time step $t$, a normal denoising operation is first performed, followed by an optional (inversion and then denoising) operation based on the value of $\lambda$. Following this logic, the notation in Algorithm 1 is correct. Here we provide a more detailed explanation of Algorithm 1.
>
> We note that $\Phi^{t}$ is denoising operation at timestep $t$ , and $\Psi^{t}$ is inversion operation. Assuming the current step is $t$, our goal is to obtain $x_{t-1}$ from $x_{t}$.
>
> - In each iteration, we first denoise $x_{t}$ under high guidance $\gamma_{1}$ to obtain a temporary $x_{t-1}$. (**Line 5**)
>
> - Determine whether zigzag operation is needed. If not, skip optimization; otherwise, perform zigzag operation. (**Line 6**)
>
> - Backtrack the temporary $x_{t-1}$ to obtain the inversion result $\tilde{x}_{t}$ under low guidance $\gamma _{2}$. (**Line 7**)
>
> - Re-denoise $\tilde{x} _{t}$ under high guidance $\gamma _{1}$ to obtain the optimized latent $x _{t-1}$. (**Line 8**)
>
> - End the current iteration and proceed to the next denoising step. (**Line 9**)
>
> We hope this further clarifies the symbols and logical meaning in the Algorithm 1.
>
> ---
>
> **Q7:** How is the guidance scale $\gamma_{1}$ set in Table 1/2?
>
> **A(7):** Due to our oversight, we only mentioned $\gamma _{1}=5.5$ in Sec 3.2 (line 235), and forgot to reiterate it in Sec 4.1. In our settings, for SDXL and SD2.1, we use $\gamma _{1}=5.5$. For DreamShaper-xl-turbo-v2, we use $\gamma _{1}=3.5$. For Hunyuan DiT, we use $\gamma _{1}=3.5$. We note that these settings are the recommended values from the official sources.

---

> ### Author Response · Authors · 2024-11-23
> **Response to Reviewer A4t2 (Part 4/4)**
>
> **Q8:** In Figure 10, how is the x-axis varied?
>
> **A(8)**:  We apologize for the inadequate description of Figure 10. It is because we use various sampling steps in this experiment, so we can oberserve the signficant improvement of Z-Sampling across various sampling steps/inference time costs .
>
> Here is a more detailed explanation:
>
> We note that in Diffusion Models, we can alter the total inference time step T to control the time cost. For example, in SDXL, the default T is 50. When we set T to 10, the time to generate one image becomes 1/5 of the original. Therefore, **by selecting different values of T, the time cost to generate one image, represented by the horizontal axis in Figure 10, also changes accordingly**.
>
> In standard sampling, the total number of denoising predictions, denoted as $T_{sd}$, is equal to T.  While in Z-Sampling,  this number, denoted as $T_{z}$, is equal to $T + 2*\lambda$.  To achieve a fair comparison in Figure 10, we set T=50 in standard sampling, meaning $T_{sd}$=50. And we set T=25 and $\lambda = \frac{T}{2}$ in Z-Sampling, mean $T_{z} = T + 2*\lambda = 25 + \frac{25}{2} * 2 = 50$.  In this scenario, the time cost to generate one image is nearly the same for Z-Sampling and standard sampling. The results in Figure 10 show that Z-Sampling consistently outperforms standard sampling.
>
> ---
>
> **Q9:** How might Z-Sampling be integrated into the training phase?
>
> **A9:** Thanks for the constructive suggestion. As we focused on sampling phase, the training phase is beyond the scope of this work. It will be very interesting to include the effect of Z-Sampling into the training phase. In future, we plan to explore training/distillating diffusion models along the path of Z-Sampling. This could leverage the improvement of Z-Sampling with no extra inference cost.
>
> Thanks for the constructive suggestion. As we focused on sampling phase, the training phase is beyond the scope of this work. It will be very interesting to include the effect of Z-Sampling into the training phase. In future, we plan to explore training/distillating diffusion models along the path of Z-Sampling. This could leverage the improvement of Z-Sampling with no extra inference cost. (copy xie sir.)
>
> ---
>
> **Q10:** The paper appears to lack analysis of the unconditional branch.
>
> **A10:** The response here is the same as in A5.
>
> ---
>
> Finally, we sincerely thanks Reviewer A4t2 for the hard work and comments again. We respectfully hope that the reviewer could reevaluate our work given the reponses addresing your main concerns. We appreciate it very much in advance.

---

> > ### Comment · Reviewer_A4t2 · 2024-11-24
> > **Thanks for your response**
> >
> > Thanks for the authors' thorough and thoughtful responses.  As most of my concerns are well addressed, I have raised my score to 5.
> > However, I still have some specific concerns that I would like to see addressed.
> >
> > **Q1: About the Oversaturation Effects**
> >
> > The authors have conducted thorough experiments on this topic, providing both qualitative and quantitative evidence, which is overall convincing. However, I would like to highlight some concerns regarding the main figure (Figure 1). In the "Counting" image, all background details seem to be lost, the green color appears unrealistic, and the overall effect does not resemble a natural picture. Similarly, in Figure 5, the "blue dog" exhibits an unnaturally vivid blue color, and the background is blurred in an unnatural manner.
> >
> > Although Figure 24 is convincing, I believe it is based on a distilled model. Distilled models typically produce different results for varying CFG scales due to differences in the distillation process.
> >
> > Hence, my questions are as follows:
> > 1. Could you provide an explanation for the unnatural appearance of the two images mentioned above? Is this effect due to SDXL or another factor?
> > 2. Could you share quantitative results using SDXL or other original T2I models? Since distilled models often exhibit distinct behavior, a comparison might help clarify these effects.
> >
> > **Q2: About the Typo**
> >
> > In Algorithm 1, I understand that the optional branch involves an inversion and denoising process. However, I noticed a potential mismatch in the comments:
> > - In line 7, the comment reads, *"Denoising by Equation (2),"* but the operation described seems related to inversion.
> > - In line 8, the comment states, *"Inversion by Equation (4),"* which appears to describe denoising instead.

---

> > > ### Author Response · Authors · 2024-11-27
> > > **Response to Reviewer A4t2's Follow-up**
> > >
> > > Thank you for your response and follow-ups! We are glad to know that our explanations and additional experiments have addressed your main concerns.
> > >
> > > We are happy to further clarify the two remaining questions.
> > >
> > > ---
> > >
> > > **Additional Q1:** Could you provide an explanation for the unnatural appearance of the two images mentioned above? Is this effect due to SDXL or another factor?
> > >
> > > **Additional A1:** Thanks for the good question. Yes, we can provide a reasonable explantion according to our empirical observation and theoretical understanding. We believe the unnatural appearance, such as mild oversaturation, that sometimes appears is a nature sideeffect of stronger guidance/prompt-image alignment. We can report similar observations on both DreamShaper and SDXL. As Figure 24 (with DreamShaper) and the additional Figure 27 (with SDXL) show, strong guidance scale can cause such sideeffect.
> > >
> > > We observe that Z-Sampling even can often mitigate this sideeffect compared with standard sampling when the guidance scale is relatively strong. For example, for the prompt "blue colored dog" in Figure 27 with SDXL, it can be observed that as the guidance scale increases, the side effects for standard sampling are more pronounced, leading to more noticeable artifacts and oversaturation. The similar observation also holds in Figure 24 with DreamShaper.
> > >
> > > Moreover, Z-Sampling can improve prompt-image alignment when the guidance scale is set to the defaulted value, where the sideeffect is often mild and acceptable. Thus, Z-Sampling may find a superior trade-off between image quality and prompt adherence.
> > >
> > > ---
> > >
> > > **Additional Q2:** Could you share quantitative results using SDXL or other original T2I models? Since distilled models often exhibit distinct behavior, a comparison might help clarify these effects.
> > >
> > > **Additional A2:**  Thanks for the constructive suggestion. In this revision, we further present the very conclusive quantitative experimental results of SD 2.1, SDXL, and DreamShaper together in the additional Table 18 (the table below). We grid search the best guidance scales for each model in terms of HPS v2 and present the results. For SDXL/SD2.1, the seached guidance range was set from 3.5 to 25.5, and for DreamShaper-xl-turbo-v2, it was set from 1.5 to 11.5. Note that previous studies commonly do not fine-tune the guidance scale hyperparameter.
> > >
> > > The conclusive quantitative results demonstrate that Z-Sampling can significant improve the best performance of all three diffusion models with various choices of the guidance scales. Moreover, the results indicate that the distilled DreamShaper with Z-Sampling can even outperform SDXL, while DreamShaper with standard sampling cannot match SDXL.
> > >
> > > | SD2.1 | HPS v2↑ | AES↑ | PickScore↑ | IR↑ | winning rate |
> > > |---------|---------|---------|---------|---------|---------|
> > > | Standard Sampling  | 0.2686   | 5.7014  | **20.3946**  | 0.2387 |  -  |
> > > | Z-Sampling  | **0.2729**  | **5.7126**  | 20.3796  | **0.2885** |  62%  |
> > >
> > > | SDXL | HPS v2↑ | AES↑ | PickScore↑ | IR↑ | winning rate |
> > > |---------|---------|---------|---------|---------|---------|
> > > | Standard Sampling  | 0.3100   | 6.0985  | 21.7238  | 0.7917 | - |
> > > | Z-Sampling  | **0.3128**   | **6.1302**  | **21.8477**  | **0.7922** |  61%  |
> > >
> > > | DreamShaper-xl-turbo-v2 | HPS v2↑ | AES↑ | PickScore↑ | IR↑ | winning rate |
> > > |---------|---------|---------|---------|---------|---------|
> > > | Standard Sampling  | 0.3016   | 5.9898  | 21.5388  | 0.6899 |  - |
> > > | Z-Sampling  | **0.3238**   | **6.1542**  | **22.1025**  | **0.9087** | 86%  |
> > >
> > > ---
> > >
> > > **Additional Q3:** About the Typo.
> > >
> > > **Additional A3:** We sincerely thank you and apologize for the typo. We note that, in Algorithm 1, the two code comments were mistakenly swapped, while the pseudocode is correct. We corrected this typo in the revision.
> > >
> > > ---
> > >
> > > We hope the additional qualitative results in Figures 24 and 27 and the comprehensive quantitative results in Table 18 can address your final concerns.

---

> > > > ### Author Response · Authors · 2024-11-30
> > > > **Look forward to your post-rebuttal feedback!**
> > > >
> > > > Dear Reviewer A4t2,
> > > >
> > > > Thanks again for your insightful suggestions and comments. Since the deadline of discussion is approaching, we are happy to provide any additional clarification that you may need.
> > > >
> > > > Based on your valuable suggestions, we provided additional results to demonstrate the distinctiveness of Z-Sampling from the high cfg effect across different models, and corrected the typo in Algorithm 1.
> > > >
> > > > We hope these new experiments and explanations have clarified the contributions and strengthened our submission. Please do not hesitate to reach out if there are any further clarifications or analyses we can provide.
> > > >
> > > > Thank you for your time and thoughtful feedback!
> > > >
> > > > Best, Authors

---

> > > > > ### Comment · Reviewer_A4t2 · 2024-12-03
> > > > > **Thanks for you response.**
> > > > >
> > > > > I appreciate the additional experiments provided by the authors, as most of my concerns have been effectively addressed. However, regarding the practical outcomes, it seems that using the default CFG scale for z-sampling may lead to some distortion in the results. I still find the quantitative performance, including the outcomes shown in Figure 1, less satisfactory. I understand that z-sampling at higher CFG scales can mitigate this issue. Therefore, I believe that when using z-sampling, a CFG scale different from the default settings in SDXL might be more appropriate. I have decided to maintain my current ratings.

---

> ### Author Response · Authors · 2024-12-03
> **Your concern is not a weakness but a feature.**
>
> We respectfully note that your final concern is not a weakness, but a feature, which does not harm the qualitative or quantitative performance of Z-Sampling. We kindly argue that your evaluation concern on the qualitative results is highly subjective, limited to very few cases, and very inconsistent with more observations, popular metrics, other three reviewers, and even previous works. In contrast, the effectiveness of Z-Sampling are widely supported and verified in terms of rich dimensions.
>
> From a qualitative perspective, in many previous works, such as Fig 3 of Diffusion-DPO [1] and Fig 2 of FreeU [2], similar oversaturation are observed. Such saturation enhancement has been widely presented and mentioned in visualization cases as an addition to some users' preference.
>
> Particularly, We note that rating the effectiveness of a method according to one user's preference to very few cases is very subjective and unreasonable. The human preference models, including HPS v2, PickScore, and Image Reward are trained with the preference of a large group of users, which reflect preference of more users, support the effectiveness of Z-Sampling under all studied CFG scales. And, even you agreed that, Figures 24 and 27 are convincing and can mitigate saturation. Table 18 even makes the advantage of Z-Sampling more convincing under various CFG scales.
>
> The dimensions of evaluation metrics include:
>
> - Higher HPS v2/PickScore/Image Reward: Better human preferences of three different large groups of users.
>
> - Higher AES: More aesthetic.
>
> - Higher GenEval performance: Better prompt adherence.
>
> - Lower FID: Better image quality/consistency with real-world images.
>
> To the best of our knowledge, no sampling method can achieve such impressive and comprehensive improvement compared to ours. If you have some candidate, please feel free to let us know. Moreover, our theoretical analysis can even further help understanding diffusion sampling and enhance our contribution beyond previous work.
>
> We gratefully thank you in advance for reconsidering the rating from a more objective perspective.
>
> Your effort will be a valuable addition to our community!
>
> Reference:
>
> [1] Wallace, Bram, et al. "Diffusion model alignment using direct preference optimization." Proceedings of the IEEE/CVF Conference on Computer Vision and Pattern Recognition. 2024.
>
> [2] Si, Chenyang, et al. "Freeu: Free lunch in diffusion u-net." Proceedings of the IEEE/CVF Conference on Computer Vision and Pattern Recognition. 2024.

---

### Official Review · Reviewer_uffk · 2024-11-05

**Soundness:** 3
**Presentation:** 3
**Contribution:** 3
**Rating:** 6
**Confidence:** 3

**Summary:**

This paper introduces Z-Sampling, a novel sampling method for diffusion models that improves image generation quality and prompt alignment. The paper investigates that if the latent variable contains the sementic information, the sampling result would be enhanced. The key insight is that the guidance gap between denoising and inversion processes can capture semantic information. Rather than doing end-to-end semantic injection, Z-Sampling accumulates semantic information step-by-step through zigzag sampling. At each step, the denoising process is followed by inversion. The authors provide a theoretical analysis showing why this approach is more effective than end-to-end injection. The paper demonstrates empirical improvements across multiple diffusion models, datasets, and evaluation metrics. The method is particularly effective for challenging prompts involving style, position, counting, and multiple objects.

**Strengths:**

1. Interesting observation and reasonable method: The paper found that the latent variable contains semantic information, and the sampling result would be enhanced. The paper does the denoising and inversion process iteratively by using different semantic scales to add the semantic information in the latent variable. The method is intuitive and reasonable.

2. Strong Theoretical Foundation: The paper provides rigorous theoretical analysis explaining why Z-Sampling works, including detailed proofs showing how semantic information accumulation differs between end-to-end and zigzag approaches.

3. Comprehensive Evaluation: The authors evaluate their method across multiple diffusion models, various datasets and different metrics. They also provide extensive ablation studies examining the impact of different parameters.

4. Practical Applicability: Z-sampling is a plug-and-play method that can enhance existing diffusion models without requiring model retraining. It can also be combined with other optimization techniques like Diffusion-DPO and AYS-Sampling for further improvements.

**Weaknesses:**

1. Large computation: Z-sampling requires additional computation due to the zigzag steps. While they show that it can achieve better results with fewer steps, the method still requires more computation than standard sampling for the same number of steps.

2. Limited to Deterministic Samplers: The method's effectiveness is reduced when used with stochastic samplers due to inversion inaccuracies. This limitation restricts its applicability to deterministic sampling methods like DDIM. Besides, the inversion such as DDIM have approximate errors. The paper should investigate how it impacts the results of some experiments.

**Questions:**

1. Could you clarify how the semantic information gain term and approximation error term interact? While equation 10 separates these terms, it's unclear how they might counteract or reinforce each other in practice.
2. Can the method be extended to Video generation or 3D generation?

---

> ### Author Response · Authors · 2024-11-23
> **Response to Reviewer uffk (Part 1/2)**
>
> We appreciate the reviewer uffk for the thorough feedback and the recognition of our contribution in both theoretical and experimental aspects. Below we address each of the reviewer uffk's concerns.
>
> ---
>
> **Q1:**  Z-Sampling incurs additional computational overhead.
>
> **A1:** We frankly admit that Z-Sampling incurs additional computational costs. We will further explore how to do Z-Sampling more efficiently in future, such as including zigzag into the training phase. However, we also note the time consumption is controllable and often very limited compared with the performance improvement. Moreover, even under the same computation overhead with less denoising steps, Z-Sampling can achieve significant improvements, according to Figure 10. This is already a powerful advantage, as the reviewer uffk also agreed.
>
> Moreover, the absolute time consumption can be often very limited in practice.
>
>   - Figure 9 demonstrates that even with a small number of steps (e.g., the first 10 steps in SDXL) employing the zigzag operation, a nearly 70% increase in winning rates can be achieved. We note that this incurs **less than a fifth of** the additional time consumption.
>
>   - In Table 1, we validate that Z-Sampling can effectively bring benefits with distillation diffusion model. Notably, the official recommended denoising steps of  DreamShaper-xl-turbo-v2 is 4, which means the additional inference steps $T_{z} - T_{sd}$ brought by Z-Sampling is **at most 6**, a very small cost, but it can increase the winning rate to 80% (Please see Figure 6).
>
> ---
>
> **Q2:**  Is Z-Sampling exclusively employed within deterministic sampling schedulers?
>
> **A2:** Yes, we observe that Z-Sampling does not perform well with stochastic samplers like Euler (A), as detailed in Appendix E.1. This is primarily due to their inability to provide precise inversion operations.
>
> We note that Z-Sampling exhibits favorable performance with mainstream ODE schedulers (such as DDIM, DPM-Solver). We respectfully contend that this underscores the versatility of our method, given that most community endeavors (such as generation acceleration[1], image editing[2]) are predicated on ODE schedulers.
>
> ---
>
> [1] Lu, Cheng, et al. "DPM-Solver++: Fast Solver for Guided Sampling of Diffusion Probabilistic Models." arXiv e-prints (2022): arXiv-2211.
>
> [2] Hertz, Amir, et al. "Prompt-to-Prompt Image Editing with Cross-Attention Control." The Eleventh International Conference on Learning Representations.

---

> ### Author Response · Authors · 2024-11-23
> **Response to Reviewer uffk (Part 2/2)**
>
> **Q3:**  The paper lacks a deeper analysis of the inversion error term
>
> **A3:** We greatly appreciate the reviewer's suggestion. We note that the theoretical analysis of the information gain term is novel and very helpful for us to design Z-Sampling. We also agreed that a deeper analysis of the error term can further provide the interesting insights and advantages of Z-Sampling beyond previous studies. We present more discussion on the error term here and in Appendix E.2/E.3 of the revision.
>
> To only study the error term  $\tau _{2}$, we eliminated the semantic term $\tau _{1}$ by setting $\delta _{\gamma} = 0$, Equation 10 is equivalently transformed as
>
> $(x_{T}-\tilde{x}_{T})^{2}=\sum _{t=1} ^{T} \alpha _{t} h _{t} ^{2} ( \underbrace{\epsilon _{\theta} ^{t}(x _{t})-\epsilon _{\theta} ^{t}(\tilde{x} _{t-1})} _{\tau _{2}(t)})$.
>
> Under this setting, we performed end-to-end and step-by-step experiments with SDXL.
>
>  |Method  | $\delta _{\gamma}$ | $PickScore↑$ |$Error _{\tau _{2}}↓$ |
> |---------|---------|---------|---------|
> | sdxl  | - | 21.6353 | - |
> | end2end   | 5.5   | 21.6485 | - |
> | step-by-step | 5.5   | **21.8477** | - |
> | end2end   | 0.0   | 18.8182 | 160.3313 |
> | step-by-step | 0.0 | **21.5257** | 0.9919 |
>
> It can be observed that without semantic gain (e.g., $\delta _{\gamma}=0$), the cumulative error  by end-to-end inversion significantly degrades image quality. In contrast, under the step-by-step setting, image quality only slightly decreases, indicating a minimal impact from $\tau _{2}$.
>
> To further illustrate, we added an additional Gaussian error term $s*error _{gs}$ as
>
> $(x_{T}-\tilde{x}_{T})^{2}=\sum _{t=1} ^{T} \alpha _{t} h _{t} ^{2} ( \underbrace{\epsilon _{\theta} ^{t}(x _{t})-\epsilon _{\theta} ^{t}(\tilde{x} _{t-1})} _{\tau _{2}(t)}+s* \frac{norm(\epsilon _{\theta} ^{t}(x _{t}))}{norm(error _{gs})} error _{gs})$
>
> | s | HPS v2↑ | AES↑ | PickScore↑ | IR↑ |
> |---------|---------|---------|---------|---------|
> | 0  | **0.2995**   | **6.1889**  | **21.5257**  | **0.5112** |
> | 0.5  | 0.2993   | 6.1502  | 21.5139  | 0.4553 |
> | 1.0   | 0.2812   | 6.0076  | 20.7824  | 0.2874 |
>
> We find that as $s$ increases, Z-Sampling performs worse. Therefore, based on the above experiments, we note that:
>
> - **An increase in the error term $\tau_{2}$ degrades the sampling effect.**
>
> - **The step-by-step approach helps reduce the error term $\tau_{2}$, mitigating this negative gain.**
>
> We provide further detailed discussions in **Appendix E.2 and E.3**.
>
> ---
>
> **Q4:**  Can Z-Sampling be extended to more generative tasks?
>
> **A4:** Thanks for the suggestion from reviewer uffk, which opens up more general potential for our method. The table below demonstrates that the mechanism of Z-Sampling remains effective in video generation.
>
> We choose AnimateDiff [1] as the baseline and test it on Chronomagic-Bench-150 [2], and we set $\gamma_{1}=7.5$ and $\gamma_{2}=0$ in Z-Sampling. We note that Z-Sampling outperforms both AnimateDiff and another train-free sampling method FreeInit [3] in UMT-FVD, UMT-Score, GPT4o-Score. More details can be found in **Table 11 in Appendix D.1**.
>
> | Method | UMT-FVD↓ | UMT-Score↑ | GPT4o-Score↑ |
> |---------|---------|---------|---------|
> |AnimateDiff|275.18|2.82|2.83|
> | FreeInit  | 268.31  | 2.82  | 2.59  |
> | Ours   | **243.26**   | **2.97**  | **2.88**  |
>
> Due to the deadline constraints, we will subsequently report the results on the 3D generation task.
>
> ---
>
> Finally, we sincerely thank the reviewer uffk’s feedback. It definitely inspires us to further improve our work and clarification for more readers.
>
> ---
>
> [1] Guo, Yuwei, et al. "AnimateDiff: Animate Your Personalized Text-to-Image Diffusion Models without Specific Tuning." ICLR, 2024.
>
> [2] Yuan, Shenghai, et al. "ChronoMagic-Bench: A Benchmark for Metamorphic Evaluation of Text-to-Time-lapse Video Generation." NeurIPS, 2023.
>
> [3] Wu, Tianxing, et al. "Freeinit: Bridging initialization gap in video diffusion models." ECCV, 2024.

---

> > ### Comment · Reviewer_uffk · 2024-11-25
> >
> > Thanks to the authors for their thoughtful response. I believe the additional theory and experiments significantly enhance the contribution of the paper. I have decided to maintain my positive score.

---

> > > ### Author Response · Authors · 2024-11-27
> > > **Response to Reviewer uffk's Follow-up**
> > >
> > > We sincerely thank you again for your suggestion and response.
> > >
> > > Following your constructive suggestions on the experiments of the error term $\tau_2$ and applying z-sampling to video diffusion models, we have significantly enhance the quality and contribution of this work, as you also agree and claim so.
> > >
> > > We would highly appreciate it if you could kindly reconsider the rating of our work given the initial score as 6 and the significantly enhanced. Your suggestion and action will both make an addition to our community.
> > >
> > > Sincerely thank you again!

---

### Author Response · Authors · 2024-11-23
**General Response to All**

We are thankful for the feedback and suggestions from all the reviewers. We are glad that the reviewers recognize our work (Z-Sampling), acknowledge our **interesting and reasonable findings** and **strong theoretical foundation** (uffk, 9XhW), **comprehensive evaluation experiments** (uffk, A4t2, BMZe, 9XhW). Additionally, its **ease of integration and practical applicability** make Z-Sampling inspiring for the community and researchers (uffk, BMZe, 9XhW).

---

Here, we would like to highlight and clarify the novelty and contributions of Z-Sampling as public comments:

- **What** signifies a better latent variable in the Diffusion Model? (Sec 3.1)

  Observing the latent's generation results under weak conditions to judge its performance under specific strong conditions.

- **How** to improve the properties of an initial latent variable to enhance generation quality? (Sec 3.2)

  Demonstrating that the cfg gap is the key component, controlling this gap can determine the effect of optimization.

- **Why** end-to-end methods is inferior to step-by-step approaches? (Sec 3.3)

  Conducting theoretical and experimental analysis to reveal the impact of semantic term $\tau _{1}$ and error terms $\tau _{2}$,  further improving the performance of Z-Sampling.

---

We duly address the reviewers' concerns in the individual responses.

Here, we summarize the supplementary experiments and contents for addressing the concerns in the revision.
- In Appendix D.4,  we demonstrate that Z-Sampling is still powerful under extremely high cfg guidance and can even mitigate oversaturation caused by strong cfg guidance.   (A4t2)

- In Appendix D.1, we presents the results of Z-Sampling on multi-step denoising and multi-step inversion settings, further demonstrating the superiority of the step-by-step approach. (BMZe)

- In Line 478-483, we provide a more detailed explanation for Figure 9. (BMZe)

- In Appendix E.2/E.3, we conduct extra experimental analysis on the error term $\tau_{2}$, which indicates that: (uffk, BMZe)

  - An increase in $\tau_{2}$ leads to negative gain.

  - A step-by-step approach can reduce the magnitude of $\tau_{2}$.

- In Appendix D.1, we demonstrate Z-Sampling remains effective in other generative tasks such as video generation. (uffk)

- In Appendix D.1, we show that Z-Sampling can also outperform recent methods for improving CFG mechanisms, such as CFG++ and SEG. (9XhW)

- In Appendix B, we discuss the relationship between Z-Sampling and other methods, such as Time-Travel, Tune-a-video, further clarifying the novelty and contributions of our work. (A4t2, BMZe)

---

Finally, we hope that these clarifications and explanations resolve the reviewer’s concerns and questions. We are more than happy to discuss if the reviewer has further questions or comments.

---

### Meta-Review · Area_Chair_7T13 · 2024-12-21

**Metareview:**

This paper proposes a sampling method for conditional diffusion models that alternates between denoising and inversion to improve the alignment of text prompts and images for T2I models. Although alternating between denoising steps and inversion steps is not new, this paper is the first to demonstrate its effectiveness for conditional generation. The Zigzag scheme is simple, effective, and generally applicable to any off-the-shelf diffusion models.
One weakness is the extra computational cost compared to standard sampling due to inversion steps. However, Zigzag can be applied to selected timestamps which makes it faster if achieving the same level of image quality and fidelity. Zigzag is also limited to deterministic samplers, and it doesn't improve image generation for stochastic samplers like DDIM.
Overall, this paper is well-motivated by interesting but somewhat evident observations. The proposed Zigzag scheme is simple, modular, and effective for image generation as well as video generation.

**Additional Comments On Reviewer Discussion:**

Reviewers were concerned with the computational complexity of the more complicated sampling scheme.
The authors have shown improved image quality with the same amount of time when Zigzag is applied to some but not all timestamps.
Reviewers were also interested in extension to other modalities beyond image generation.
The authors have included more experiments with video generation.
One reviewer raised a major concern about observed artifacts of oversaturation with the proposed method.
The authors have convinced the reviewers that the proposed method actually mitigates the issue of oversaturation with large guidance scales.
One reviewer suggested more recent work that should be compared to.
The authors have included additional comparisons to these methods which further shows the effectiveness of the proposed method.

---

### Decision · Program_Chairs · 2025-01-22

Accept (Poster)